# LEARNING INPUT-AGNOSTIC MANIPULATION DIRECTIONS IN STYLEGAN WITH TEXT GUIDANCE

**Yoonjeon Kim[1], Hyunsu Kim[2], Junho Kim[2], Yunjey Choi[2], Eunho Yang[1,3]***

Korea Advanced Institute of Science and Technology (KAIST)[1], NAVER AI LAB[2], AITRICS[3]
yoonkim313@kaist.ac.kr hyunsu1125.kim@navercorp.com
jhkim.ai@navercorp.com yunjey.choi@navercorp.com eunhoy@kaist.ac.kr

## ABSTRACT

With the advantages of fast inference and human-friendly flexible manipulation, image-agnostic style manipulation via text guidance enables new applications that were not previously available. The state-of-the-art text-guided image-agnostic manipulation method embeds the representation of each channel of StyleGAN independently in the Contrastive Language-Image Pre-training (CLIP) space, and provides it in the form of a *Dictionary* to quickly find out the channel-wise manipulation direction during inference time. However, in this paper we argue that this dictionary which is constructed by controlling single channel individually is limited to accommodate the versatility of text guidance since the collective and interactive relation among multiple channels are not considered. Indeed, we show that it fails to discover a large portion of manipulation directions that can be found by existing methods, which manually manipulates latent space without texts. To alleviate this issue, we propose a novel method **Multi2One** that *learns* a *Dictionary*, whose entry corresponds to the representation of a single channel, by taking into account the manipulation effect coming from the interaction with multiple other channels. We demonstrate that our strategy resolves the inability of previous methods in finding diverse known directions from unsupervised methods and unknown directions from random text while maintaining the real-time inference speed and disentanglement ability.

## 1 INTRODUCTION

Wide range of generative models including adversarial networks (Goodfellow et al., 2014; Karras et al., 2018; 2019; 2020b; Kim et al., 2022; Kim & Ha, 2021; Karras et al., 2021), diffusion models (Dhariwal & Nichol, 2021), and auto-regressive models (Dosovitskiy et al., 2020; Chang et al., 2022) have demonstrated notable ability to generate a high-resolution image that is hardly distinguishable from real images. Among these powerful models, style-based GAN models (Karras et al., 2019; 2020b) are equipped with a unique latent space which enables style and content mixing of given images, manipulation of local regions (Wu et al., 2021), and interpolation between different class of images (Sauer et al., 2022).

In this paper, we focus on the image manipulation based on the pre-trained StyleGAN, considering the unique advantages mentioned above and its popularity. Based on the steerability in the latent space of StyleGAN, researchers have put tremendous effort on finding a direction that causes semantically equivalent change to the entire samples of image. In this work, we refer to such latent direction as *global direction*. Unlike *local direction* which is a sample-wise traversal direction found by iterative optimization using a single image (Local Basis (Choi et al., 2021) and Latent Optimization of StyleCLIP (Patashnik et al., 2021)), global direction allows fast inference and is applicable to any images once found using supervised (Jahanian et al., 2019), unsupervised (Shen & Zhou, 2021; Wang & Ponce, 2021; Härkönen et al., 2020; Voynov & Babenko, 2020), or text-guided methods (Global Mapper & `GlobalDirection`[1] of StyleCLIP (Patashnik et al., 2021)).

---

[1] In order to distinguish it from global direction, which means finding input agnostic directions, we express the method proposed in StyleCLIP in this way

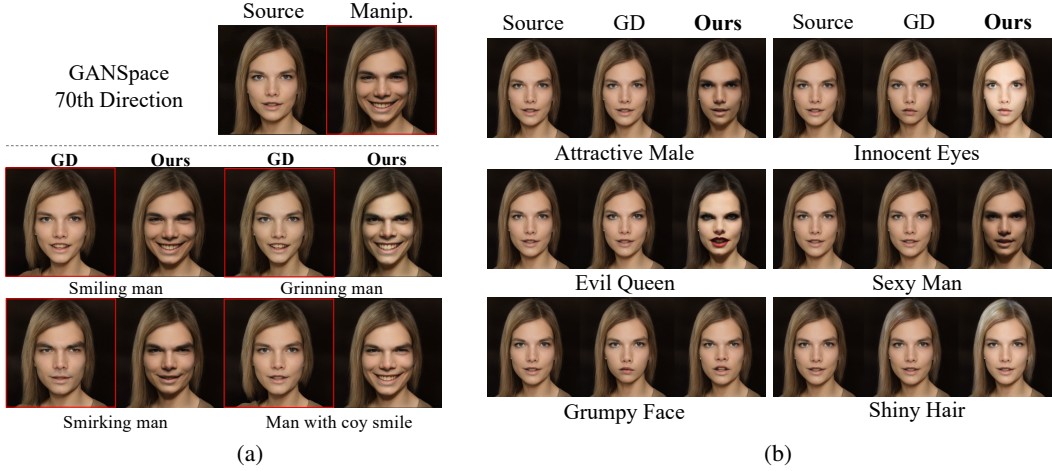

Figure 1: (a) Manipulation by the 70-th direction from GANspace generates 'a man with wide smile'. `GlobalDirection` (GD), highlighted in red, fails to reproduce similar result even when provided with various text guidances. (b) Manipulation results by *randomly selected text*, demonstrating that GD has insufficient manipulation ability. Same number of channels are manipulated in both methods.

Among them, the text-guided methods have a unique advantage in that they can naturally provide the flexibility of manipulation through the diversity of the given driving text without human supervision to discover the direction in the latent space. However, in this paper we argue that contrary to this common belief on text-guidance, the standard method (Patashnik et al., 2021) for text-based StyleGAN manipulation surprisingly fails to even find the manipulation directions that are known to be found in unsupervised approaches (Härkönen et al., 2020; Shen & Zhou, 2021) (see Fig. 1(a) for examples). In addition, we also show that this standard method does not properly perform manipulation on a large number of randomly selected texts (see Fig. 1(b) for examples). We hypothesize that the failure is due to the naïve approach that only considers a change of image caused by *a single channel* in StyleSpace, neglecting diverse directions that are visible only when manipulating multiple channels as a whole.

In order to address these issues, we propose a novel method, named **Multi2One**, of *learning* a *Dictionary* that can manipulate multiple channels corresponding to a given text. However, here since there is no paired ground truth of text and manipulation direction corresponding to the text, we embed the directions found by existing unsupervised methods into the CLIP space and learn a dictionary to reproduce them in the CLIP space. Note that this has more meaning than simply reproducing the known directions derived by unsupervised methods. As the dictionary learns the relationship between channels in StyleSpace and CLIP space, we can find manipulations that could not be found with unsupervised methods using diverse text inputs.

Through extensive experiments, we confirm that contrary to the state-of-the-arts method (Patashnik et al., 2021) which explicitly encoded every single channel, our multi-channel based strategy not only excels in reconstruction of unsupervised directions but also in discovery of text-guided directions.

## 2  RELATED WORK

**Style-based Generators**   Generators of style-based models (Karras et al., 2019; 2020b;a; 2021) are built upon the progressive structure (Karras et al., 2018) that generates images of higher resolution in deeper blocks. The popularity of StyleGAN structure that has been employed in numerous number of researches comes from its ability to generate high-fidelity images, transfer styles to other images, and manipulate images in the latent spaces using inversion methods (Zhu et al., 2020; Roich et al., 2021; Tov et al., 2021; Collins et al., 2020). The latent spaces of StyleGAN used for manipulation are intermediate space $\mathcal{W}$ and StyleSpace $\mathcal{S}$ (Wu et al., 2021).

**Unsupervised Global Directions**   Image-agnostic directions are latent vectors that create semantically equivalent shift when applied to the latent space of StyleGANs. In order to find such directions, SeFa (Shen & Zhou, 2021) performs PCA on the first weight that comes after intermediate space $\mathcal{W}$ in pre-trained StyleGAN, deducing the principal components as the global directions. On the other hand, GANspace (Härkönen et al., 2020) relies on the randomly sampled latent codes in $\mathcal{W}$ and

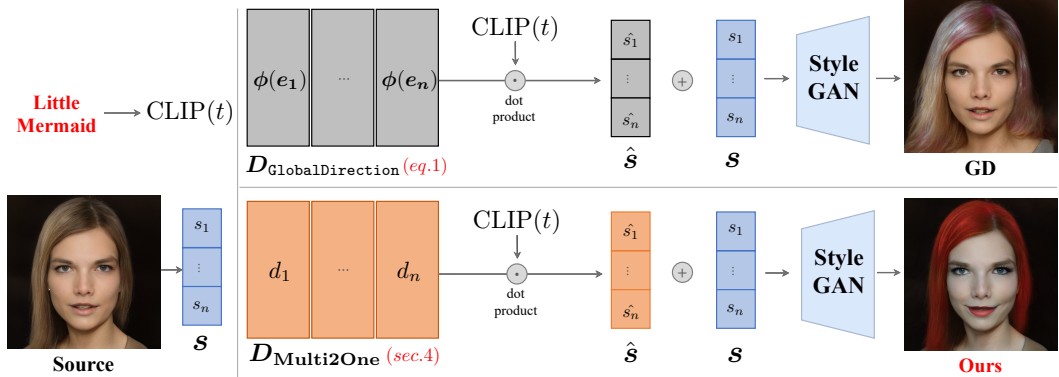

Figure 2: A diagram depicting the framework of dictionary-based image manipulation via text guidance. Our method, **Multi2One**, is differentiated from the previous methods in that it learns the dictionary for the text input. The proposed novel dictionary allows more flexible and expansive discovery of the manipulation direction $\hat{s}$ with better result. $\phi(\cdot)$ in $D_{\texttt{GlobalDirection}}$ is an abbreviation of $\phi_{\text{CLIP}}(\cdot)$ of Eq. (1). The dictionary learning process of **Multi2One** to create $D_{\textbf{Multi2One}}$ is illustrated in Sec. 4.

the eigenvectors from the latent codes proved to be global directions that share an image-agnostic modification ability.

**Text-guided Image Manipulations**   Most of the text-guided image manipulation methods aim to find a local direction (Kocasari et al., 2022; Xia et al., 2021; Patashnik et al., 2021) which is an image-specific direction that applies to a single sample of image. Methods that find local directions using text guidance could be found in various models including GAN, Diffusion (Kim & Ye, 2021; Nichol et al., 2021; Avrahami et al., 2022; Liu et al., 2021), and Vision transformers (Chang et al., 2022). Two unique approaches for finding a global direction using text guidance in StyleGAN are *Global Mapper* and GlobalDirection of StyleCLIP (Patashnik et al., 2021). Global Mapper finds an image-invariant direction for a single text by optimizing a fully connected layer. However, this method requires 10 hours of training time for every single text, making it less popular than GlobalDirection. On the other hand, GlobalDirection method offers a real-time manipulation in inference time using a dictionary-based framework that is applicable to any input image and text.

## 3   LIMITED COVERAGE OF STYLECLIP GLOBALDIRECTION METHOD

In this section, we briefly review the GlobalDirection of StyleCLIP (Patashnik et al., 2021), which performs StyleGAN-based image manipulation using text guidance (Sec. 3.1). Then, we provide our key motivation that this state-of-the-art text-guided manipulation method is surprisingly insufficient to fully utilize the manipulative power of StyleGAN (Sec. 3.2).

### 3.1   TEXT-GUIDED STYLEGAN IMAGE MANIPULATION

**StyleSpace of StyleGAN**   The generators of StyleGAN family (Karras et al., 2019; 2020a;b) have a number of latent spaces: $\mathcal{Z}$, $\mathcal{W}$, $\mathcal{W}+$, and $\mathcal{S}$. The original latent space is $\mathcal{Z}$, which is typically the standard normal distribution. The generator transforms an input noise $z \sim N(0, I)$ into intermediate latent spaces $\mathcal{W}$ (Karras et al., 2019), $\mathcal{W}+$ (Abdal et al., 2019), and $\mathcal{S}$ (Wu et al., 2021), sequentially. Recent study (Wu et al., 2021) shows that StyleSpace $\mathcal{S}$ is the most disentangled such that it can change a distinct visual attribute in a localized way. The number of style channels in $\mathcal{S}$ is 6048 excluding toRGB channels for StyleGAN-ADA (resolution $1024^2$), and recent methods (Kocasari et al., 2022; Patashnik et al., 2021; Wu et al., 2021) modify the values of the channels to edit an image. Our method also adopts StyleSpace and we use $n$ to denote the total number of channels in $\mathcal{S}$ (that is, $\boldsymbol{s} = [s_1, s_2, \cdots, s_n]^T \in \mathcal{S}$ and $s_i$ is a single parameter). For the convenience of explanation, the pre-trained generator $G$ is re-defined with respect to StyleSpace as $X = G_s(\boldsymbol{s})$; $X$ is a generated image. The goal of StyleGAN-based image manipulation via text is to find a direction $\hat{s} = [\hat{s}_1, \hat{s}_2, ..., \cdots, \hat{s}_n]^T$ in StyleSpace $\mathcal{S}$ which generates an image $X_{\text{edited}} = G_s(\boldsymbol{s} + \hat{s})$ suitable for the provided text guidance $t$. Note that $\boldsymbol{s}$ is the inverted style vector of image $X$, found via StyleGAN inversion methods such as Alaluf et al. (2021); Roich et al. (2021); Tov et al. (2021) , used for manipulation purpose in most cases. The main benefit of considering StyleSpace $\mathcal{S}$ in image manipulation is that it does not change the undesired regions of the image when modifying a small number of style channels by its well disentangled property.

Table 1: Measurement of CLIP similarity score $cos(\cdot, \cdot)$ ($\uparrow$) between the manipulated image and the CLIP representation $\phi_{\text{CLIP}}(\cdot)$ of unsupervised direction $\boldsymbol{\alpha}$.

|  | $cos(\phi_{\text{CLIP}}(\boldsymbol{\alpha}), X_{\text{unsup}})$ | $cos(\phi_{\text{CLIP}}(\boldsymbol{\alpha}), X_{\text{GD}})$ |
|---|---|---|
| $\boldsymbol{\alpha}^{(10)}$ | **0.3295** | 0.2701 |
| $\boldsymbol{\alpha}^{(30)}$ | **0.3558** | 0.3319 |
| $\boldsymbol{\alpha}^{(50)}$ | **0.3687** | 0.2924 |

**Text-guided Manipulation by StyleCLIP**  The `GlobalDirection` (Patashnik et al., 2021) is a representative method of text-driven image manipulation that provides an input-agnostic direction with a level of disentanglement. Intuitively, it computes the similarity between the input text and each style channel in the CLIP space (Radford et al., 2021) to find the channels that should be modified given the text. In order to compute the similarity between the text and style channel, both should be encoded into CLIP. While the text guidance $t$ is trivially encoded via the text encoder of CLIP as $\text{CLIP}(t) \in \mathbb{R}^p$, *style* channels in StyleSpace $\mathcal{S}$ need additional pre-processing for the embedding. `GlobalDirection` proposes to embed the manipulation effect of $i$-th *style* channel using the following mapping from the StyleSpace $\mathcal{S}$ to CLIP space:

$$\phi_{\text{CLIP}}(\boldsymbol{e}_i) := \mathbb{E}_{\boldsymbol{s} \in \mathcal{S}}\Big[\text{CLIP}\big(G_s(\boldsymbol{s} + \boldsymbol{e}_i)\big) - \text{CLIP}\big(G_s(\boldsymbol{s} - \boldsymbol{e}_i)\big)\Big] \qquad (1)$$

where the manipulation vector $\boldsymbol{e}_i \in \mathbb{R}^n$ is a zero-vector except for the $i$-th entry. Adding the manipulation vector $\boldsymbol{e}_i$ to the original style vector $\boldsymbol{s}$ indicates that only the $i$-th channel among $n$ channels in StyleSpace is manipulated. Note that $\phi_{\text{CLIP}}(\boldsymbol{e}_i)$ is also $p$-dimensional since the CLIP encoder maps images generated by $G_s(\cdot)$ into a $p$-dimensional CLIP space. The above mapping in Eq. (1) is enumerated across all $n$ channels in StyleSpace $\mathcal{S}$ to create a dictionary $D_{\texttt{GlobalDirection}} := [\phi_{\text{CLIP}}(\boldsymbol{e}_1), \phi_{\text{CLIP}}(\boldsymbol{e}_2), \cdots, \phi_{\text{CLIP}}(\boldsymbol{e}_n)] \in \mathbb{R}^{p \times n}$.

Finally, with this dictionary, the manipulation direction $\hat{\boldsymbol{s}} \in \mathbb{R}^n$ by `GlobalDirection` is given as the similarity score measured by the following equation:

$$\hat{\boldsymbol{s}} = D_{\texttt{GlobalDirection}}^T \text{CLIP}(t). \qquad (2)$$

This overall manipulation procedure is visualized in Fig. 2.

In the following section (Sec. 3.2), we propose the evidence to the hypothesis that the single-channel encoding strategy with $\phi_{\text{CLIP}}(\boldsymbol{e}_i)$ to create the dictionary $D$ is the major bottleneck causing limited coverage issues that StyleCLIP suffers from.

## 3.2 COVERAGE ANALYSIS OF STYLECLIP GLOBALDIRECTION METHOD

In this section, we experimentally verify that `GlobalDirection` has limited ability to find manipulation directions, as described in Fig. 1 of the introduction. Toward this we show two empirical findings, which corresponds to Fig. 1(a) and Fig. 1(b), respectively.

First, we show that many edits using unsupervised methods (Härkönen et al., 2020; Shen & Zhou, 2021) cannot be recovered by `GlobalDirection`. For example, in Fig. 1(a), we can observe that applying the 70-th GANspace direction manipulates the source image to become a man with wide smile, showing that pre-trained StyleGAN itself is capable of such manipulation. However, `GlobalDirection` (GD) in Fig. 1(a) constantly fails to recover the same editing effect of the unsupervised direction, despite the variation on the text input such as 'smiling man', 'grinning man', 'smirking man', and 'man with coy smile'.

More quantitatively, we provide Tab. 1 to show that `GlobalDirection` (Patashnik et al., 2021) cannot effectively recover the directions found by unsupervised methods (Härkönen et al., 2020; Shen & Zhou, 2021). Scores in the table are the CLIP similarity between $\phi_{\text{CLIP}}(\boldsymbol{\alpha}) = \mathbb{E}_{\boldsymbol{s} \in \mathcal{S}}[\text{CLIP}\big(G_s(\boldsymbol{s} + \boldsymbol{\alpha})\big) - \text{CLIP}\big(G_s(\boldsymbol{s} - \boldsymbol{\alpha})\big)]$ and the modified image created by either `GlobalDirection` ($X_{\text{GD}}$; the construction of $X_{\text{GD}}$ is explained below)[2] or unsupervised methods ($X_{\text{unsup}}$). Note that $\phi_{\text{CLIP}}(\boldsymbol{\alpha})$ encodes the direction $\boldsymbol{\alpha}$ into the CLIP embedding space as `GlobalDirection` does for a single

---

[2]We performed the experiment here using the `GlobalDirection` method and test examples provided on the StyleCLIP official site: https://github.com/orpatashnik/StyleCLIP

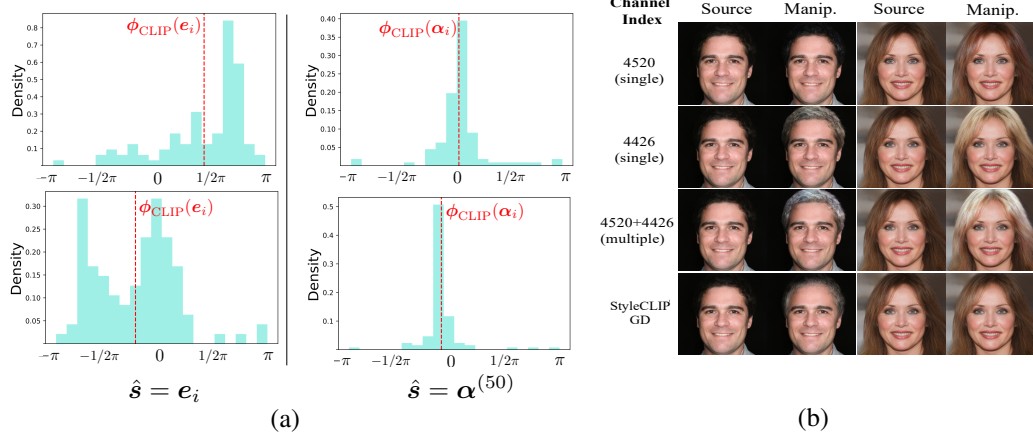

Figure 3: (a) Histogram of $\text{CLIP}\big(G_s(s+\hat{s})\big) - \text{CLIP}\big(G_s(s-\hat{s})\big)$ using single-channel manipulation $e_i$ or multiple channel manipulation $\alpha^{(50)}$. (b) Manipulation results using channel indices 4520, 4426. `GlobalDirection` (denoted as StyleCLIP GD) manipulated with text 'white hair'.

channel. From the table, we can confirm that `GlobalDirection` does not recover the similarity in CLIP space that the unsupervised methods have.

Here we illustrate the details on how we generate $X_{\text{unsup}}$ and $X_{\text{GD}}$ for Tab. 1. First, the source image $X = G_s(s)$ is manipulated by each unsupervised direction $\alpha \in \mathbb{R}^n$ into $X_{\text{unsup}} = G_s(s + \alpha)$. Instead of naïvely using the direction $\alpha \in \mathbb{R}^n$, we only use the channels whose magnitude is the $k$-largest among $n$ channels in $\alpha$ to reduce the entanglement that naturally comes with directions found by unsupervised methods (please refer to Appendix C to find the effect of using limited number of channels). We denote such sparsified directions as $\alpha^{(k)} \in \mathbb{R}^n$ and use them in the experiment of Tab. 1. Then, to generate $X_{\text{GD}} = G_s(s + \hat{\alpha})$, we compute $\hat{\alpha} = D^T_{\texttt{GlobalDirection}} \text{CLIP}(t) \in \mathbb{R}^n$ as in Eq. (2). However, since there is no text corresponding to $\alpha$ and our objective in this experiment is to find a direction $\hat{\alpha}$ that may recover $X_{\text{unsup}}$ by `GlobalDirection`, we replace $\text{CLIP}(t)$ with $\phi_{\text{CLIP}}(\alpha^{(k)}) = \mathbb{E}_{s \in \mathcal{S}}[\text{CLIP}\big(G_s(s+\alpha^{(k)})\big) - \text{CLIP}\big(G_s(s-\alpha^{(k)})\big)]$. Since $\phi_{\text{CLIP}}(\alpha^{(k)})$ indicates the manipulation effect of $\alpha^{(k)}$ encoded into CLIP space, it is used as a substitute for the CLIP-encoded text guidance, $\text{CLIP}(t)$. Finally, we compute the similarity score for 30 instances of source images and 1024 directions from SeFa and GANspace whose average is reported in Tab. 1.

Secondly in Fig. 1(b), we provide several instances where the `GlobalDirection` of StyleCLIP fails to manipulate an image given randomly selected text. For additional examples on the failure cases of the standard method, please refer to Appendix A where we provide the details on how we choose random texts and manipulated images using extensive list of random texts.

Regarding the phenomena found in the above experiments, we hypothesize the crude assumption of `GlobalDirection` - manipulation solely based on single channel could fully represent the manipulative ability of StyleGAN - leads to the limited coverage issue. To disprove this assumption of `GlobalDirection`, in Fig. 3(a), we demonstrate that single-channel manipulation framework of `GlobalDirection` (using $e_i$) leads to improper construction of $\phi_{\text{CLIP}}(e_i)$ in the dictionary $D_{\texttt{GlobalDirection}}$. Then in Fig. 3(b), we show an example where editing two channels has a completely different effect from manipulating each of the channels individually, which proves the need for multi-channel manipulation.

In Fig. 3(a), we show that $\phi_{\text{CLIP}}(e_i) = \mathbb{E}\big[\text{CLIP}\big(G_s(s+e_i)\big) - \text{CLIP}\big(G_s(s-e_i)\big)\big]$, which forms the dictionary $D_{\texttt{GlobalDirection}}$, is not a robust representative of $i$-th style channel since there exists sample-wise inconsistency in the calculation of $\text{CLIP}\big(G_s(s+e_i)\big) - \text{CLIP}\big(G_s(s-e_i)\big)$. On the left side of Fig. 3(a), the distribution of $\text{CLIP}\big(G_s(s+e_i)\big) - \text{CLIP}\big(G_s(s-e_i)\big)$ and its average over the samples, $\phi_{\text{CLIP}}(e_i)$, are shown. As the samples show inconsistent angles over polar coordinate[3] when modified by a single-channel manipulation $e_i$, we conclude that $\phi_{\text{CLIP}}(e_i)$ may not be a trust-worthy representative of $i$-th channel in StyleSpace. Thereby the basic assumption held by `GlobalDirection` which expects a single channel manipulation using $e_i$ on any images to be semantically consistent in CLIP space, in fact, has numerous counter-examples showing inconsistency.

---

[3]A more detailed explanation on representing the samples as directions for an angle histogram is deferred to Appendix B due to space constraints.

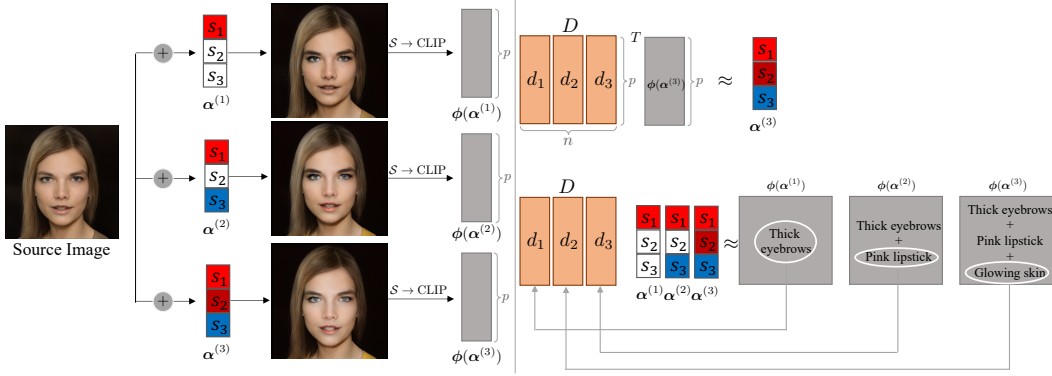

Figure 4: A concept diagram of our method in learning the *dictionary D*. $\phi(\cdot)$ is an abbreviation of $\phi_{\text{CLIP}}(\cdot)$. $\mathcal{S} \rightarrow$ CLIP represents a mapping from StyleSpace to CLIP.

On the other hand, it is observable on the right side of Fig. 3(a), that manipulation by unsupervised direction $\boldsymbol{\alpha}^{(50)}$, shows much consistent results.

Moreover, in Fig. 3(b), we show an example where single channel alone has completely different meaning from the manipulation by the combination of multiple channels. This supports our claim that the interaction between multiple channels should be considered, rather than individually encoding single channel each. In Fig. 3(b), manipulation by 4520-th or 4426-th do not show "completely white hair" while the combination of the two shows white hair. Since `GlobalDirection` fails to learn that 4520-th also has a role for creating white hair, it fails to successfully modify an image when text guidance is given as 'white hair'.

## 4 DICTIONARY LEARNING FOR SPARSE REPRESENTATION OF CLIP SPACE

As discussed in Sec. 3.2, each style channel rather than the whole may not be meaningfully embedded in CLIP space. To address this issue, we propose a dictionary learning approach to find a robust representation of *styles* that could be possibly a set of channels in StyleSpace $\mathcal{S}$.

Ideally, given a text input $t$ and its CLIP encoding $\text{CLIP}(t) \in \mathbb{R}^p$, we would like to construct a dictionary $D \in \mathbb{R}^{p \times n}$ which finds a manipulation direction $\hat{s} = D^T \text{CLIP}(t) \in \mathbb{R}^n$ as in Eq. (2). The direction $\hat{s}$ is expected to manipulate an image into $G_s(s + \hat{s})$ which faithfully represents $t$. Let us for now assume that the 'ground truth' manipulation direction $\hat{s}_t \in \mathbb{R}^n$ corresponding to the driving text $t$ is known. Then our estimated direction $\hat{s} = D^T \text{CLIP}(t)$ should generate an image $X_{\text{edited}} = G_s(s + D^T \text{CLIP}(t))$ close to $G_s(s + \hat{s}_t)$. This leads to the objective

$$\underset{D}{\text{minimize}} \left\| D^T \text{CLIP}(t) - \hat{s}_t \right\|_2^2 \tag{3}$$

which we name *ideal* dictionary learning problem since the ground truth direction $\hat{s}_t$ for a driving text $t$ is unknown. To avoid this unrealistic assumption, we substitute $\hat{s}_t$ with the known directions $\boldsymbol{\alpha} \in \mathbb{R}^n$ derived from unsupervised methods (Härkönen et al., 2020; Shen & Zhou, 2021). $\text{CLIP}(t)$ is also replaced by $\phi_{\text{CLIP}}(\boldsymbol{\alpha})$ (see Eq. (1) for the definition of $\phi_{\text{CLIP}}(\cdot)$) since we would like to use $\boldsymbol{\alpha}$ itself as a driving guidance rather than finding out corresponding text $t$ that requires labor-intensive labeling. Then, Eq. (3) becomes

$$\underset{D}{\text{minimize}} \sum_{\forall \boldsymbol{\alpha}} \left\| D^T \phi_{\text{CLIP}}(\boldsymbol{\alpha}) - \boldsymbol{\alpha} \right\|_2^2. \tag{4}$$

We construct the set of known directions $\boldsymbol{\alpha}$ in Eq. (4) by using all 512 directions found by GANspace and directions with top 80 eigenvalues out of 512 from SeFa[4] (Härkönen et al., 2020; Shen & Zhou, 2021). Instead of naïvely using the unsupervised direction $\boldsymbol{\alpha} \in \mathbb{R}^n$, we prune the channels in directions to activate only $k$ channels among $n$ channels, zeroing out the other $n - k$ channels that have small magnitudes. Such pruning is represented by $\boldsymbol{\alpha}^{(k)} \in \mathbb{R}^n$ where we use $k = 10, 30, 50$ in this paper (the effect of $k$ is described in Tab. 5 in the appendix). The reason for pruning the number of activated channels is that unsupervised methods show instability in terms of maintaining

---

[4]This is based on the observation that the change in image is not significant for the directions with lower eigenvalues in SeFa.

Table 2: Mean Squared Error (Eq. (6); ↓) measured in StyleSpace. We manipulate total of 100 channels using both `GlobalDirection` and **Ours**. Note that $\boldsymbol{\alpha}$ is $\boldsymbol{\alpha}^{(100)}$ in this metric.

| | Global Direction | Ours |
|---|---|---|
| $\mathbf{MSE}(\boldsymbol{\alpha}, \hat{\boldsymbol{\alpha}})$ | $0.0315_{\pm 0.0037}$ | $\mathbf{0.0057}_{\pm 0.0034}$ |
| $\mathbf{MSE}(\boldsymbol{\alpha}, \hat{\boldsymbol{\alpha}}\vert_{\neq 0})$ | $0.0267_{\pm 0.00114}$ | $\mathbf{0.0005}_{\pm 0.00001}$ |
| $\mathbf{MSE}(\boldsymbol{\alpha}, \hat{\boldsymbol{\alpha}}\vert_{=0})$ | $0.006218_{\pm 0.00658}$ | $\mathbf{0.00486}_{\pm 0.00363}$ |

the original image due to the entanglement (please refer to Appendix C for examples of entanglement without pruning) and we circumvent such issue by constraining the direction $\boldsymbol{\alpha} \in \mathbb{R}^n$ to have limited effect on only $k$ channels.

To help close the gap between Eq. (3) and Eq. (4), we introduce an additional loss term $\|D\boldsymbol{\alpha}^{(k)} - \boldsymbol{\phi}_{\text{CLIP}}(\boldsymbol{\alpha}^{(k)})\|_2^2$ to update the dictionary $D \in \mathbb{R}^{p \times n}$. This additional term comes from the observation in Sec. 3.2 that multi-channel manipulation should be considered. Unlike `GlobalDirection` which directly encodes $\boldsymbol{\phi}_{\text{CLIP}}(\boldsymbol{e}_i)$ for all $n$ channels in StyleSpace, we aim to learn the multi-channel manipulation by $\boldsymbol{\alpha}^{(k)}$ whose editing effect is mapped from StyleSpace to CLIP space by $\boldsymbol{\phi}_{\text{CLIP}}(\cdot)$. The main obstacle in this objective is identifying which channel is responsible for which part of the change induced by total of $k$ channels in the given direction $\boldsymbol{\alpha}^{(k)}$. Here, we emphasize that augmenting the same direction $\boldsymbol{\alpha}$ into $\boldsymbol{\alpha}^{(10)}, \boldsymbol{\alpha}^{(30)}, \boldsymbol{\alpha}^{(50)}$ can encourage the dictionary learning process to identify a disentangled editing effect of each channel which will be explained further below.

Based on the concept figure in Fig. 4, we explain how the additional loss term encourages disentangled learning of dictionary $D$. We present a simplified version where $k = 1, 2, 3$, creating three augmentations $\boldsymbol{\alpha}^{(1)}, \boldsymbol{\alpha}^{(2)}, \boldsymbol{\alpha}^{(3)}$ for the direction $\boldsymbol{\alpha}$. Suppose manipulation using $\boldsymbol{\alpha}^{(3)} = [s_1, s_2, s_3]^T$ modifies the source image to have 1) thick eyebrows, 2) pink lipstick and 3) glowing skin. Further suppose manipulation by $\boldsymbol{\alpha}^{(2)} = [s_1, 0, s_3]^T$ modifies 1) thick eyebrows and 2) pink lipstick. Finally, manipulation using $\boldsymbol{\alpha}^{(1)} = [s_1, 0, 0]^T$ modifies 1) thick eyebrows. If the loss term $\|D\boldsymbol{\alpha}^{(k)} - \boldsymbol{\phi}_{\text{CLIP}}(\boldsymbol{\alpha}^{(k)})\|_2^2$ is perfectly optimized to satisfy $D\boldsymbol{\alpha}^{(k)} = \boldsymbol{\phi}_{\text{CLIP}}(\boldsymbol{\alpha}^{(k)})$, then $s_1 d_1 \approx \text{CLIP(thick eyebrows)}, s_1 d_1 + s_3 d_3 \approx \text{CLIP(thick eyebrows \& pink lipsticks)}$ and $s_1 d_1 + s_2 d_2 + s_3 d_3 \approx \text{CLIP(thick eyebrows \& pink lipsticks \& glowing skin)}$. Therefore, the linear combination on the sparsified direction helps to specifically link 'thick eyebrows' to $d_1$ 'pink lipstick' to $d_3$, and 'glowing skin' to $d_2$. Hence our method may have greatly disentangled and diverse directions coming from the enriched coverage over manipulation in StyleGAN, thanks to the dictionary learning based on multi-channel manipulation. Moreover, even though our $\boldsymbol{\alpha}$ comes from the unsupervised methods, our method learns the individual role of $s_1$, $s_2$ and $s_3$ from the single unsupervised direction $\boldsymbol{\alpha} = [s_1, s_2, s_3]^T$. This leads to greatly diverse and disentangled results compared to unsupervised methods which will be proved in Tab. 3.

Finally, combining these two ingredients together yields our dictionary learning problem by considering all known directions $\boldsymbol{\alpha}^{(k)}$:

$$\underset{D}{\text{minimize}} \sum_{\forall \boldsymbol{\alpha}^{(k)}} \left\| D^T \boldsymbol{\phi}_{\text{CLIP}}(\boldsymbol{\alpha}^{(k)}) - \boldsymbol{\alpha}^{(k)} \right\|_2^2 + \lambda \cdot \left\| D\boldsymbol{\alpha}^{(k)} - \boldsymbol{\phi}_{\text{CLIP}}(\boldsymbol{\alpha}^{(k)}) \right\|_2^2 \qquad (5)$$

where $\lambda$ is the tunable hyper-parameter and is set as 0.01 in all our experiments below.

## 5 EXPERIMENTS

We evaluate the proposed method on StyleGAN2 (Karras et al., 2020a). We used StyleGAN2-ADA pretrained on multiple datasets, including FFHQ (Karras et al., 2019), LSUN (Yu et al., 2015) Car, Church, AFHQ (Choi et al., 2020) Dog, Cat. In this section, we show the widened StyleGAN manipulation coverage of our method compared to a text-guided method, `GlobalDirection` of StyleCLIP (Patashnik et al., 2021) mainly on FFHQ pre-trained model. For additional details in experimental design, please refer to Appendix E.

**Quantitative Results**   We measure the superiority of our method in terms of recovering the unsupervised directions $\boldsymbol{\alpha}$ and text guidance $t$. The metrics are defined in StyleSpace $\mathcal{S}$ to evaluate over

Table 3: Cosine similarity $R_{\mathrm{CLIP}}$ (Eq. (7); ↑) measured in CLIP Space. For unsupervised method, which does not depend on text guidance, we choose the text-related unsupervised direction $\boldsymbol{\alpha}$ that has the highest similarity value of $cos(\boldsymbol{\phi}_{\mathrm{CLIP}}(\boldsymbol{\alpha}), \mathrm{CLIP}(t))$ given text $t$. We also present the similarity value on our method where hyper-parameter $\lambda$ is 0 for ablation study on the second term of Eq. (5).

|  | Methods | Unsupervised | Global Direction | Ours ($\lambda = 0$) | Ours ($\lambda = 0.01$) |
|---|---|---|---|---|---|
| Manip. Channels | **10** | 0.219 | 0.224 | 0.235 | **0.242** |
|  | **30** | 0.221 | 0.228 | 0.240 | **0.249** |
|  | **50** | 0.223 | 0.232 | 0.243 | **0.251** |
|  | **100** | 0.225 | 0.237 | 0.248 | **0.254** |

unsupervised directions and in CLIP space for evaluation over text guidance. **Multi2One** recover the unsupervised direction with maximum manipulative ability and minimum entanglement, for demonstration of which we define two metrics as follows.

- **StyleSpace metric for *unsupervised directions* (Tab. 2):** We employ the following metric:

$$\mathbf{MSE}(\boldsymbol{\alpha}, \hat{\boldsymbol{\alpha}}) := \mathbb{E}_{\boldsymbol{\alpha} \sim \mathbb{A}}\big[\|\hat{\boldsymbol{\alpha}} - \boldsymbol{\alpha}\|_2^2\big] \tag{6}$$

  where $\boldsymbol{\alpha}$ is from a set of unsupervised directions $\mathbb{A}$. Note that the ground truth direction $\boldsymbol{\alpha}$ in this metric is $\boldsymbol{\alpha}^{(100)}$, where top-100 channels from unsupervised directions are modified while other $n - 100$ channels are zero entries. Therefore the reconstruction of such direction $\boldsymbol{\alpha}^{(100)}$, given by $\hat{\boldsymbol{\alpha}} = D^T \boldsymbol{\phi}_{\mathrm{CLIP}}(\boldsymbol{\alpha}^{(100)})$ is expected to have 100 *style* channels whose values are similar to that of non-zero entries in $\boldsymbol{\alpha}^{(100)}$ while the other $n - 100$ *styles* are close to 0. $\mathbf{MSE}(\boldsymbol{\alpha}, \hat{\boldsymbol{\alpha}}|_{\neq 0})$ denotes the difference between $\boldsymbol{\alpha}$ and $\hat{\boldsymbol{\alpha}}$ measured between the 100 non-zero *style* channels that represents the ability of the given dictionary $D$ to find the precise manipulation direction. On the other hand, $\mathbf{MSE}(\boldsymbol{\alpha}, \hat{\boldsymbol{\alpha}}|_{=0})$ is the distance between $\boldsymbol{\alpha}$ and $\hat{\boldsymbol{\alpha}}$ measured within $n - 100$ zero-entries. This represents the level of disentanglement since zero-entries are not expected to be modified in $\hat{\boldsymbol{\alpha}}$.

- **CLIP space metric for *text guidance* (Tab. 3):** We measure the similarity between edited image and the text guidance in CLIP space as follows:

$$\mathbf{R}_{\mathrm{CLIP}}(t, X_{\mathrm{edited}}) := \mathbb{E}_{t \sim \mathbb{T}}\big[cos\big(\mathrm{CLIP}(t), \mathrm{CLIP}(X_{\mathrm{edited}})\big)\big]. \tag{7}$$

  We denote the set of text prompts that describes the pre-trained domain of StyleGAN, namely FFHQ as $\mathbb{T}$ whose construction is described in Appendix A.

Tab. 2 shows that **Multi2One** consistently outperforms `GlobalDirection` in terms of MSE score between the unsupervised direction $\boldsymbol{\alpha}$ and the estimated direction $\hat{\boldsymbol{\alpha}}$. Especially, note that **Multi2One** have lower $\mathbf{MSE}(\boldsymbol{\alpha}, \hat{\boldsymbol{\alpha}}|_{=0})$ than `GlobalDirection`, proving that our method shows less entanglement.

Moreover, Tab. 3 shows that the cosine similarity between the given text guidance and the manipulated image is always the highest using our method. Most importantly, we emphasize that our method could find manipulation directions that could not have been found by unsupervised methods. We prove this claim using 57 text prompts ($t \sim \mathbb{T}$) as a guidance to manipulate the images. Since we aim to find the unsupervised direction that could edit the image to become $t$, we select the $\boldsymbol{\alpha}$ whose similarity $cos(\boldsymbol{\phi}_{\mathrm{CLIP}}(\boldsymbol{\alpha}), \mathrm{CLIP}(t))$ is the largest then manipulate the image to produce $X_{\mathrm{unsup}} = G_s(\boldsymbol{s} + \boldsymbol{\alpha})$. In Tab. 3, **Multi2One** scores better $\mathbf{R}_{\mathrm{CLIP}}(t, X_{\mathrm{edited}})$ compared to unsupervised methods. Therefore, we claim that our method successfully learns to find expansive editing directions which could not be found by unsupervised methods given diverse text inputs. Furthermore, we report $\mathbf{R}_{\mathrm{CLIP}}(t, X_{\mathrm{edited}})$ score with ablation on the second term of Eq. (5) in Tab. 3. For visual examples see Appendix G.

**Qualitative Results** We demonstrate the effectiveness of our method, compared to the state-of-the-arts method, `GlobalDirection`. We conducted all experiments using the pretrained models and the precomputed set of CLIP embeddings for StyleCLIP, that are readily provided for FFHQ, AFHQ Cat and AFHQ Dog. We computed the CLIP embeddings of the StyleSpace of LSUN Church and Car using the same configuration with `GlobalDirection`. All the results are compared under same condition, where manipulation strength and disentanglement levels are set to be equal. More specifically, we modify same number of channels for both methods and the magnitude of change[5] in StyleSpace is always fixed as 10.

---

[5]The original paper StyleCLIP refers to the magnitude of change in StyleSpace as $\alpha$ and the disentanglement level as $\beta$ which we substitute by number of channel that are changed.

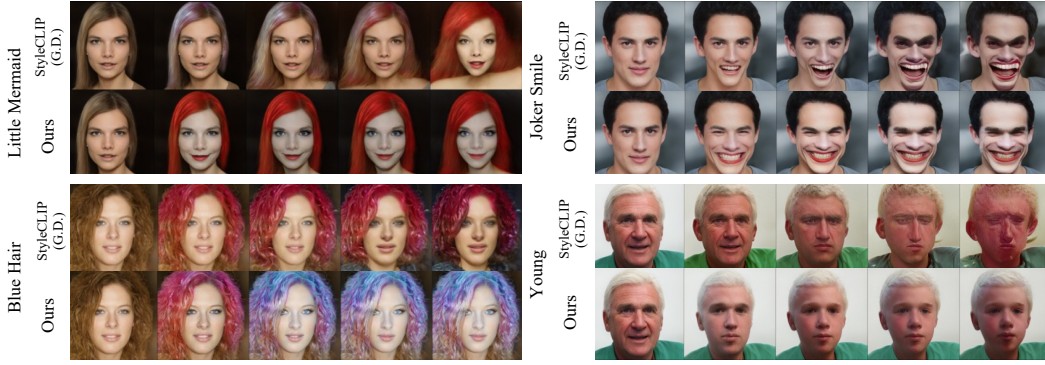

Figure 5: Manipulation results using FFHQ-pretrained StyleGAN2-ADA via text guidance on our method and StyleCLIP `GlobalDirection`. The samples are manipulated with increasing number of channels that are modified leading to larger changes.

The manipulated images show that `GlobalDirection` fails to represent some of the most basic text such as 'Young' for FFHQ. On the other hand, ours successfully manipulates the images not only on the basic text guidances but also on advanced semantics such as 'Little mermaid' and 'Joker smile' for FFHQ. We emphasize that despite the dictionary of our method is learned from the known directions in unsupervised approaches (Shen & Zhou, 2021; Härkönen et al., 2020), the manipulation results show that our learned dictionary could adapt to previously unseen combination of semantics such as red hair, pale skin, and big eyes to represent 'Little Mermaid' and unnatural smiles with red lipstick and pale face to represent 'Joker smile'.

Here we show that our method successfully discovers a direction that does not exist in unsupervised direction. To be more specific, we manipulate the image using an unsupervised direction $\alpha$ whose CLIP representation, $\phi_{\text{CLIP}}(\alpha)$, is the most similar with the text 'Little Mermaid' and 'Joker smile'. Then from the manipulated image, we observe that the manipulation result is hardly plausible compared to the successful results in Fig. 5. Based on this examples, we emphasize that even though the dictionary learning process relies on the unsupervised methods, **Multi2One** demonstrate wider variety of manipulation results using the versatility of text.

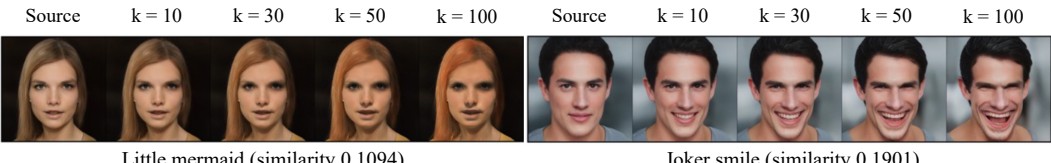

Figure 6: **Manipulation by the direction with highest cosine similarity**. We find the unsupervised direction with highest similarity given text 'Little mermaid' and 'Joker smile'. Then we manipulate the image using top-$k$ channels from the found direction.

We show additional results of our method compared to `GlobalDirection` based on AFHQ and LSUN dataset in Appendix D due to space constraints.

## 6 CONCLUSION

Text-guided image manipulation has been largely preferred over unsupervised methods in finding manipulation directions due to its ability to find wide variety of directions corresponding to flexible text input. However, we have investigated that the state-of-the-art method of text-guided image manipulation has limited coverage over possible editing directions using StyleGAN and this problem rises from a simple assumption that a single-channel manipulation could fully represent the manipulation ability of StyleGAN. To overcome this issue, we have proposed a dictionary learning framework that embeds the interactive and collective editing effect based on modifying multiple style channels into CLIP space jointly. Our method have been proven to be superior over other text-guided image manipulation methods in terms of manipulative ability and disentanglement.

## REPRODUCIBILITY STATEMENT

Our code is open sourced at here.

## ACKNOWLEDGEMENTS

This work was supported by Institute of Information & communications Technology Planning & Evaluation (IITP) grant funded by the Korea government(MSIT) (No.2019-0-00075, Artificial Intelligence Graduate School Program(KAIST)). This project is also supported by KAIST-NAVER Hypercreative AI Center.

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

## A  MORE EXAMPLES ON FIG. 1(B)

In this section, we show the extensive results on the limited coverage of `GlobalDirection` on wide variety of text guidance.

The experiment is conducted on FFHQ-pretrained StyleGAN-ADA, and the text guidance are the adjective-noun pairs extracted from Visual Semantic Ontology (Jou et al., 2015) from which extract the nouns that describe human face. For example, the adjective-noun pairs contain human face descriptions such as 'Handsome smile', 'Bad hair', and 'Stupid face'. The number of such adjective-noun pair amounts to 57 instances.

We manipulate the original image on the leftmost side using `GlobalDirection` and **Multi2One** using the same set of text description and equal configuration for both methods (manipulated channels 5, manipulation strength 5). The result in Fig. 7 shows that StyleCLIP `GlobalDirection` constantly fails to represent the given text guidance faithfully, manifesting its limited capacity in exploiting the flexibility of text hence the need for **Multi2One** to overcome the coverage issue.

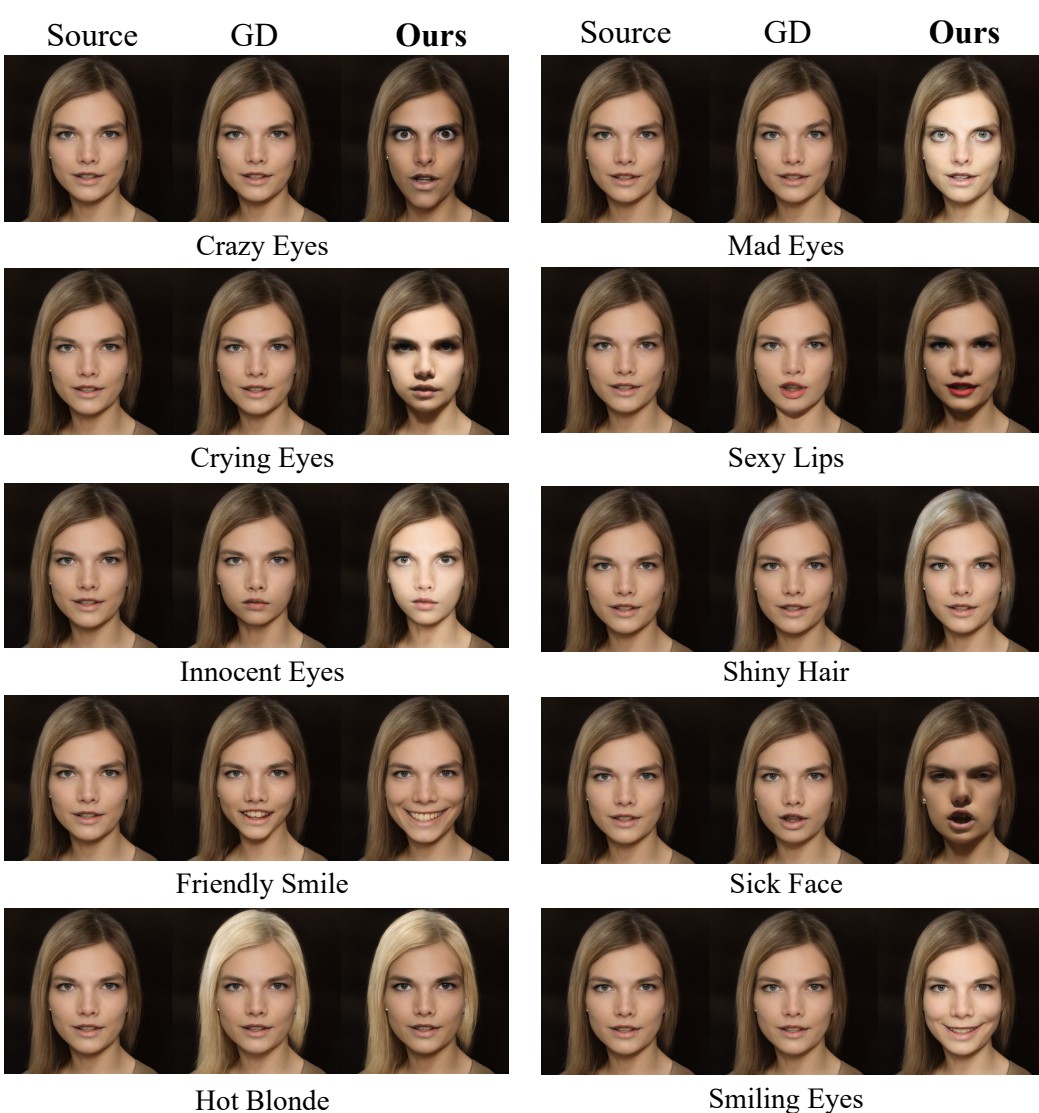

Figure 7: Manipulated images using randomly selected text from Visual Semantic Ontology (VSO). We show manipulated image via `GlobalDirection` (denoted as GD) and **Multi2One**.

## B    More Details on Fig. 3(a)

Both `GlobalDirection` and **Multi2One** rely on multiple samples of images to encode the averaged difference caused by manipulating a sample into positive and negative direction. The difference is that unlike `GlobalDirection` which relies on a single channel manipulation, **Multi2One** encode change of image caused by image-agnostic direction found by unsupervised methods (Härkönen et al., 2020; Shen & Zhou, 2021). Single channel manipulation is not image-agnostic since the change caused by moving a single channel may not be applicable to certain source images used as a sample. For unconditional FFHQ pretrained StyleGAN-ADA, we have observed that a style channel which is known to make a person's hair grey fails to modify half of the samples. On the other hand, the image-agnostic directions found by unsupervised methods hardly fails to manipulate any given image to become older with grey hair. As a consequence, the CLIP embeddings of difference caused by single channel manipulation may greatly vary sample by sample while image-agnostic directions show consistent direction of CLIP embeddings due to the successful manipulation on any given sample of images.

We visualize the direction of each sample when a single channel is modified, and when 100 or 200 channels from unsupervised directions are modified. More specifically, for a single channel manipulation of `GlobalDirection` the samples are $E\big(G_s(s + s_i)\big) - E\big(G_s(s - s_i)\big)$ and for multiple channel manipulation, the samples are $E\big(G_s(s + \alpha)\big) - E\big(G_s(s - \alpha)\big)$ where $s \sim \mathcal{S}$.

The visualization is conducted in a similar manner with the work of Wang & Isola (2020). First we embed the $p = 512$ dimensional CLIP vectors into 2-dimension using t-SNE (Van der Maaten & Hinton, 2008). We denote each sample as $(a_i, b_i)$ for $i = 1, 2, \cdots, 100$ and the averaged direction of interest $\mathbb{E}\left[E\big(G_s(s + s_i)\big) - E\big(G_s(s - s_i)\big)\right]$ in 2-dimensional space is $(\overline{a}, \overline{b})$. The angles are derived as $\theta_i = \arctan2(b_i, a_i)$ for all samples which range between $[-\pi, +\pi]$. The histogram of 100 angles are plotted with the angle of averaged difference in $\mathbb{R}^2$ with the angle $\arctan2(\overline{b}, \overline{a})$ is emphasized as a red line in Fig. 3(a).

## C    EXAMPLES OF ENTANGLEMENT USING UNSUPERVISED METHODS

In Fig. 8, we demonstrate the effect of using limited number of channels among all non-zero channels in unsupervised direction $\boldsymbol{\alpha} \in \mathbb{R}^n$.

Since StyleGAN generator consists of progressiveGAN structure where the resolution of generated image progressively increases, it is possible to classify the generator structure into three blocks: coarse, medium and fine. We apply the unsupervised directions on each of the blocks. Moreover, by filtering out to only use $k$ manipulated channels, we show that the source image shows less drastic change. By manipulating all channels, which is shown at the rightmost side of the figure, the source image is drastically changed showing entanglement.

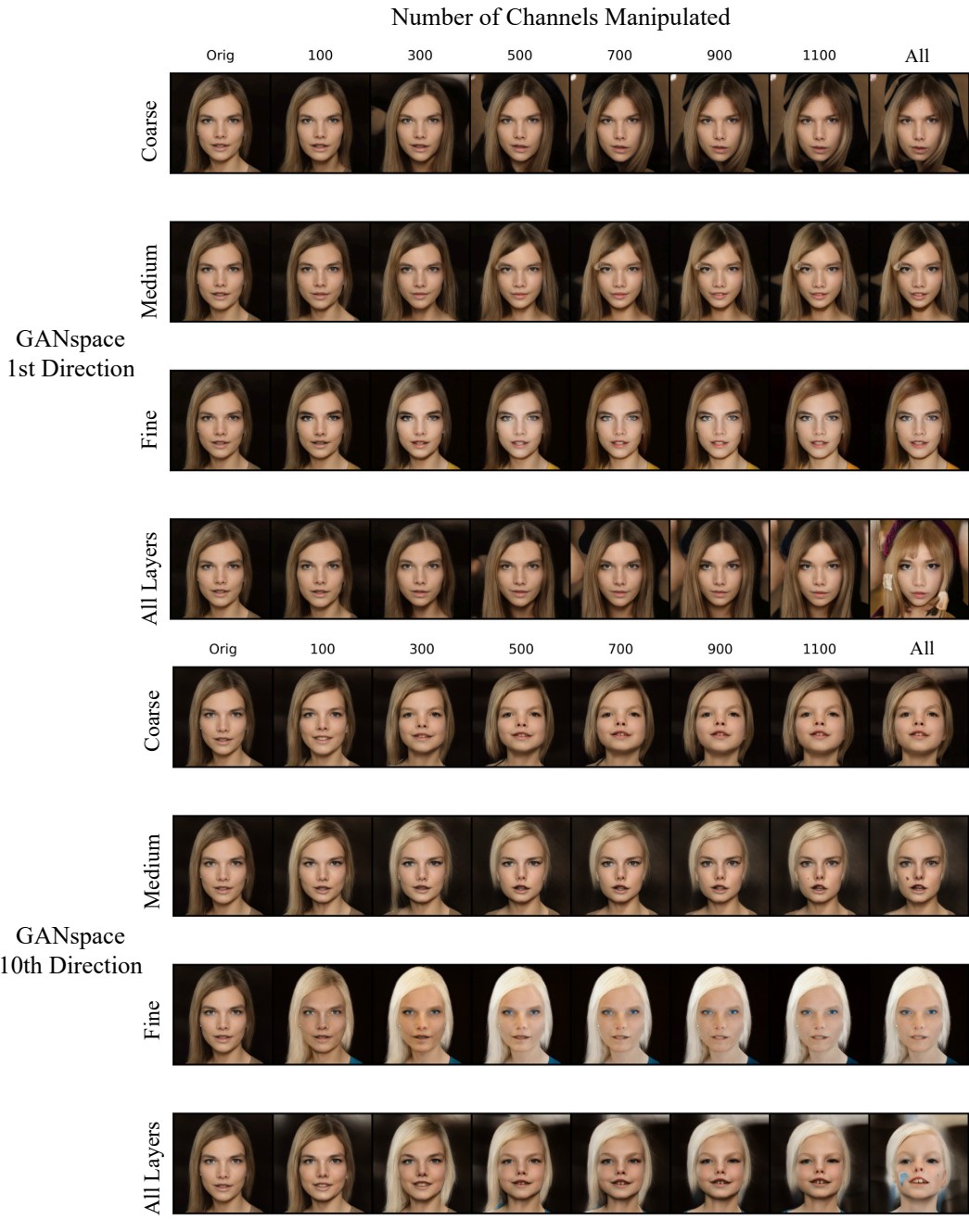

Figure 8: The manipulation results using two directions from GANSpace shows that the source identity is impaired when modifying all 18 layers of $\mathcal{W}$. Moreover, limiting the number of manipulated channels for each coarse, medium, and fine layers leads to significantly better results in terms of disentanglement.

## D    MORE EXPERIMENTAL EXAMPLES FOR ALL DATASET

In this section, we provide more experimental results of **Multi2One** with FFHQ, AFHQ, and LSUN.

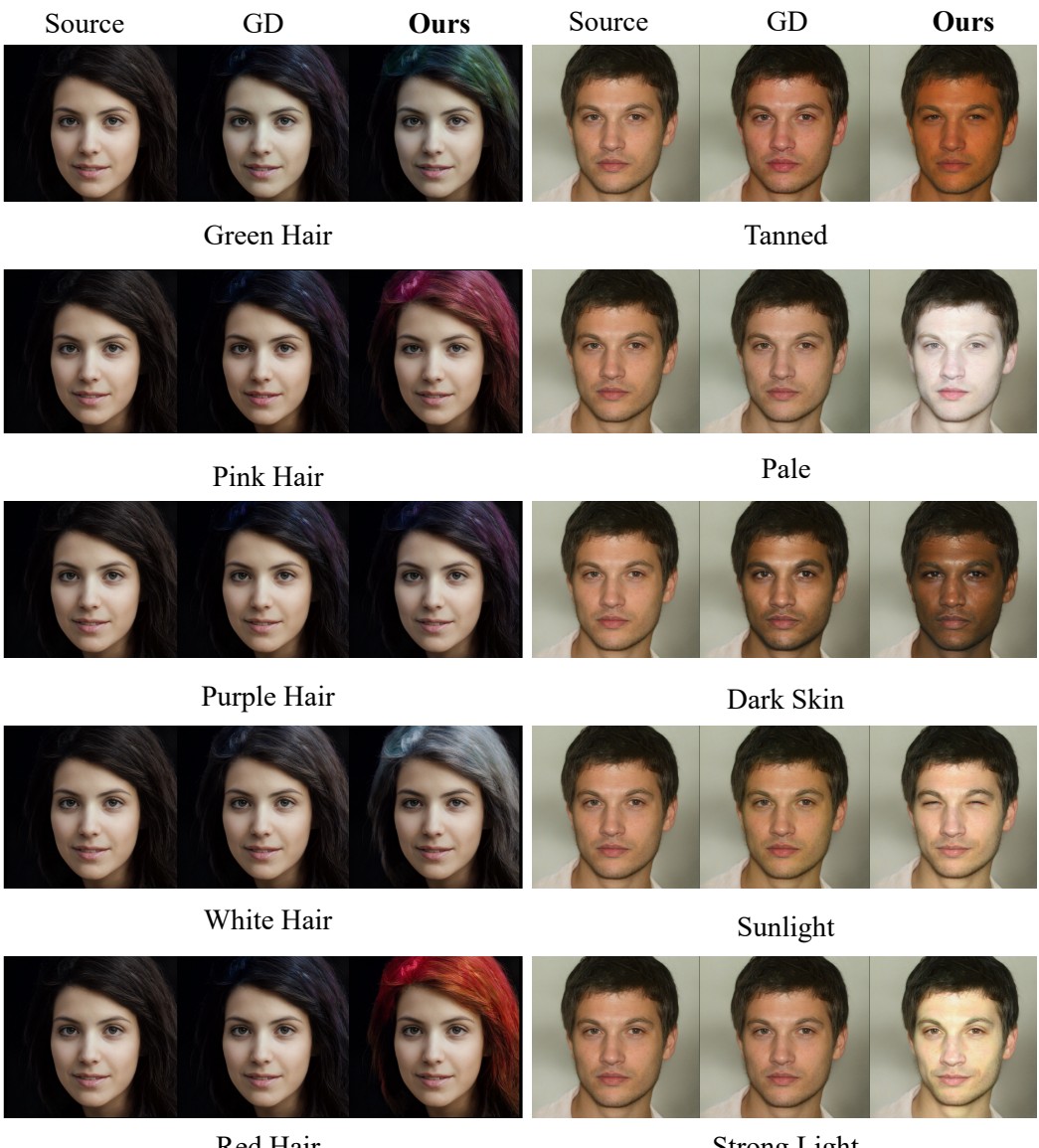

Figure 9: Additional manipulation results on FFHQ-pretrained StyleGAN-ADA. Equal configuration for both `GlobalDirection` (denoted as GD) and **Multi2One**.

Fig. 10 shows the manipulation results on **Multi2One** and StyleCLIP `GlobalDirection` using 100, 500, 1000 channels. In addition, the manipulation results without channel pruning strategy is shown as well. Using over 500 channels is required for large structural change of the object in source image. However, StyleCLIP `GlobalDirection` shows signs of severe image collapse before creating an image that perfectly satisfies the given text guidance. For example, an image at the first row with 100 channels manipulated do not seem 'young' enough and the extent of manipulation is not improved with using more channels since the image collapse and entanglement becomes a dominating factor. On the other hand, **Multi2One** shows consistently reasonable and disentangled images even when using all channels in STYLESPACE.

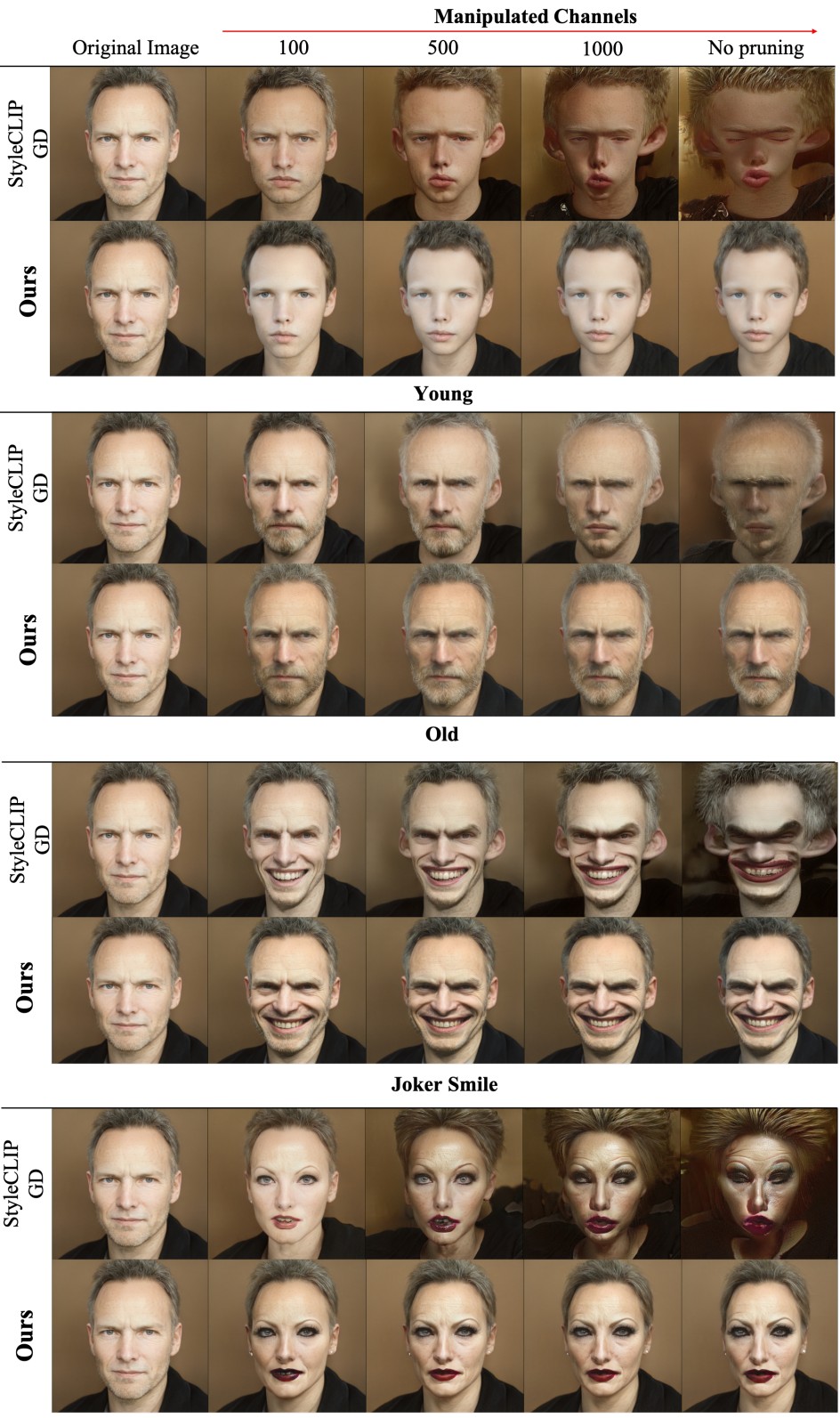

Figure 10: Manipulation results showing the disentanglement of **Multi2One** when using 100, 500, 1000 channels, and without pruning channels. StyleCLIP `GlobalDirection` manifests image collapse when using over 500 channels while **Multi2One** shows robustness against entanglement.

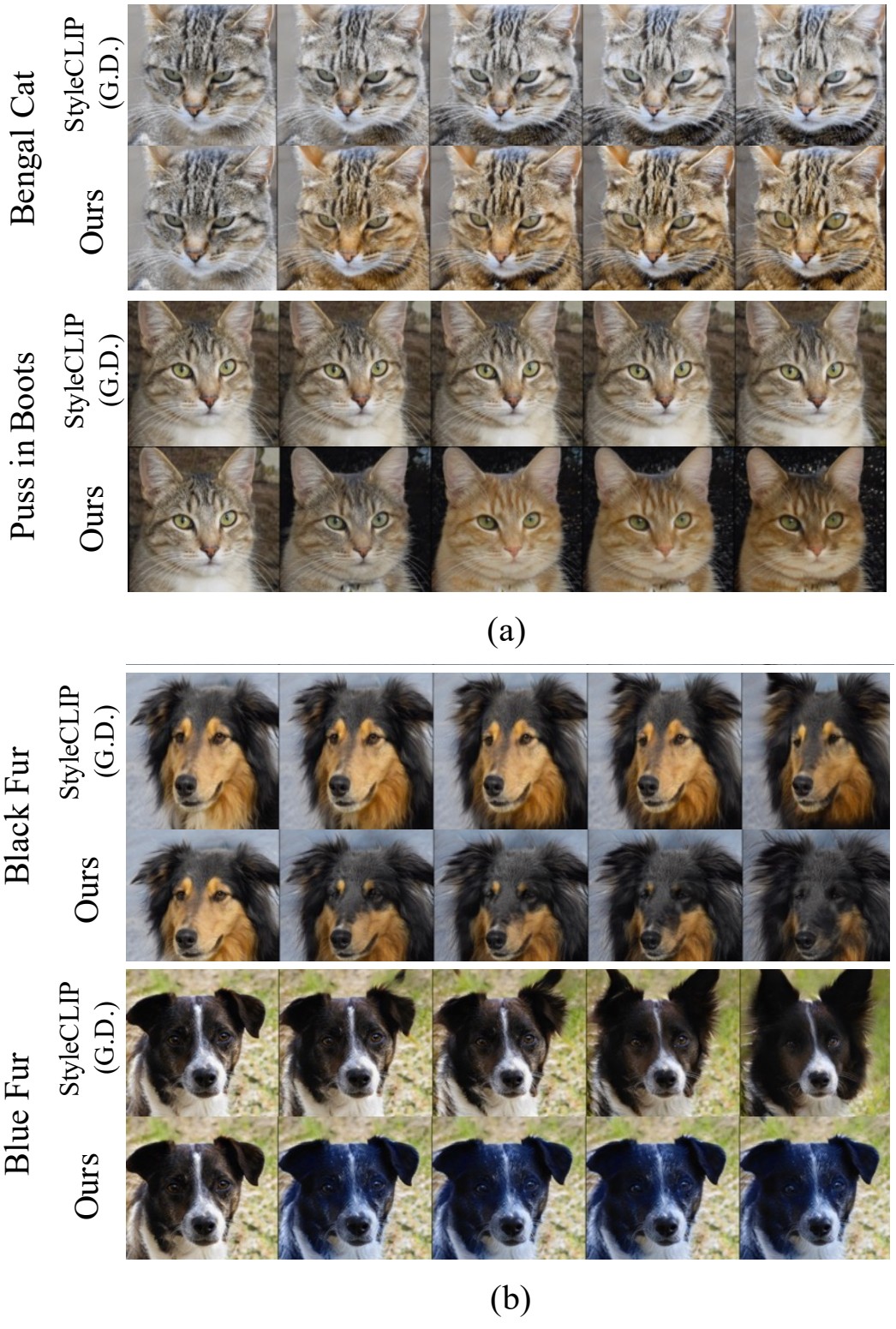

Figure 11: Manipulation results using AFHQ cat and AFHQ dog pretrained StyleGAN2-ADA via text guidance on our method **Multi2One** and StyleCLIP `GlobalDirection`. The samples are manipulated with increasing number of channels that are modified leading to larger changes.

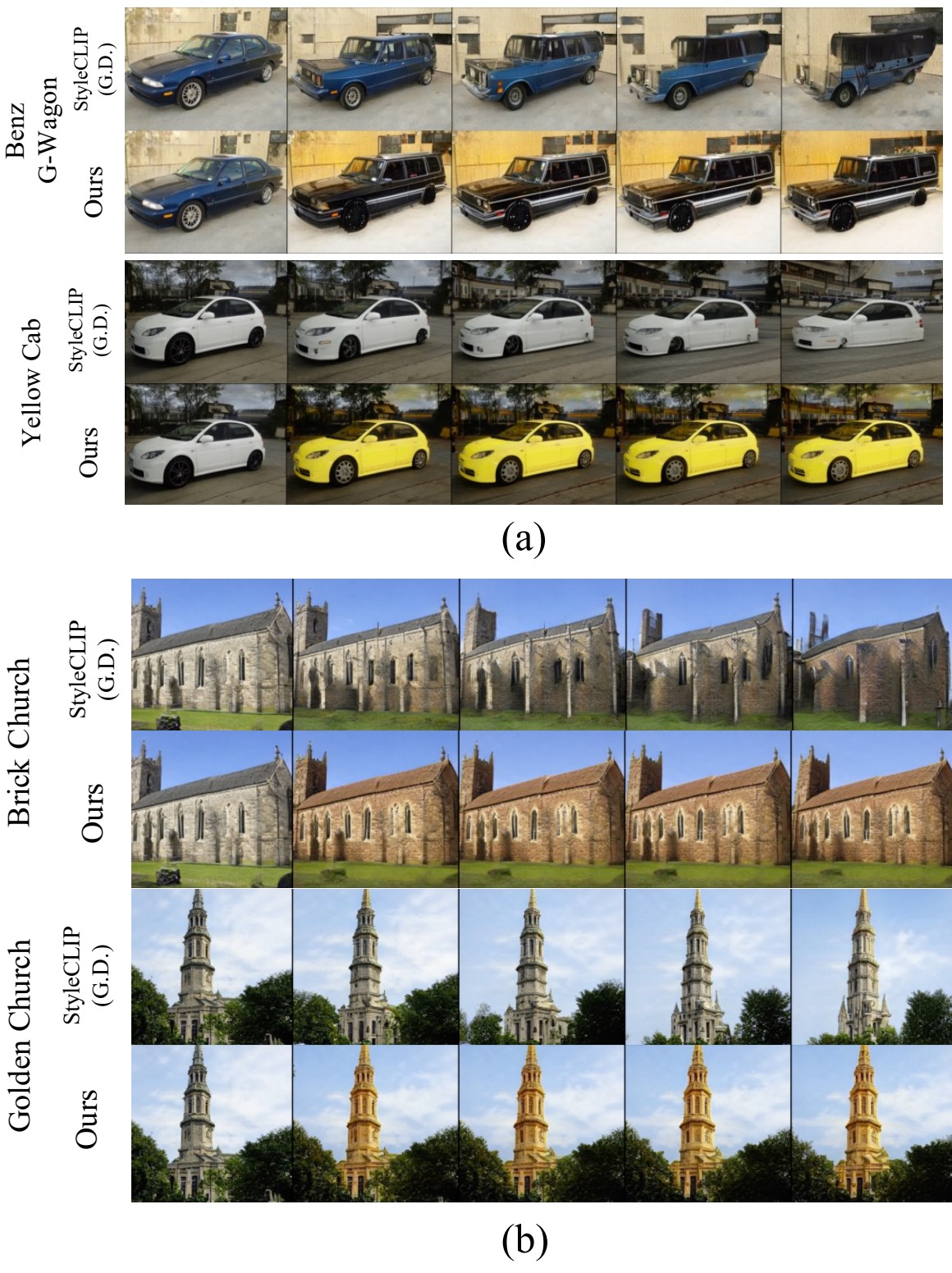

Figure 12: Manipulation results using LSUN car and LSUN church pretrained StyleGAN2-ADA via text guidance on our method **Multi2One** and StyleCLIP `GlobalDirection`. The samples are manipulated with increasing number of channels that are modified leading to larger changes.

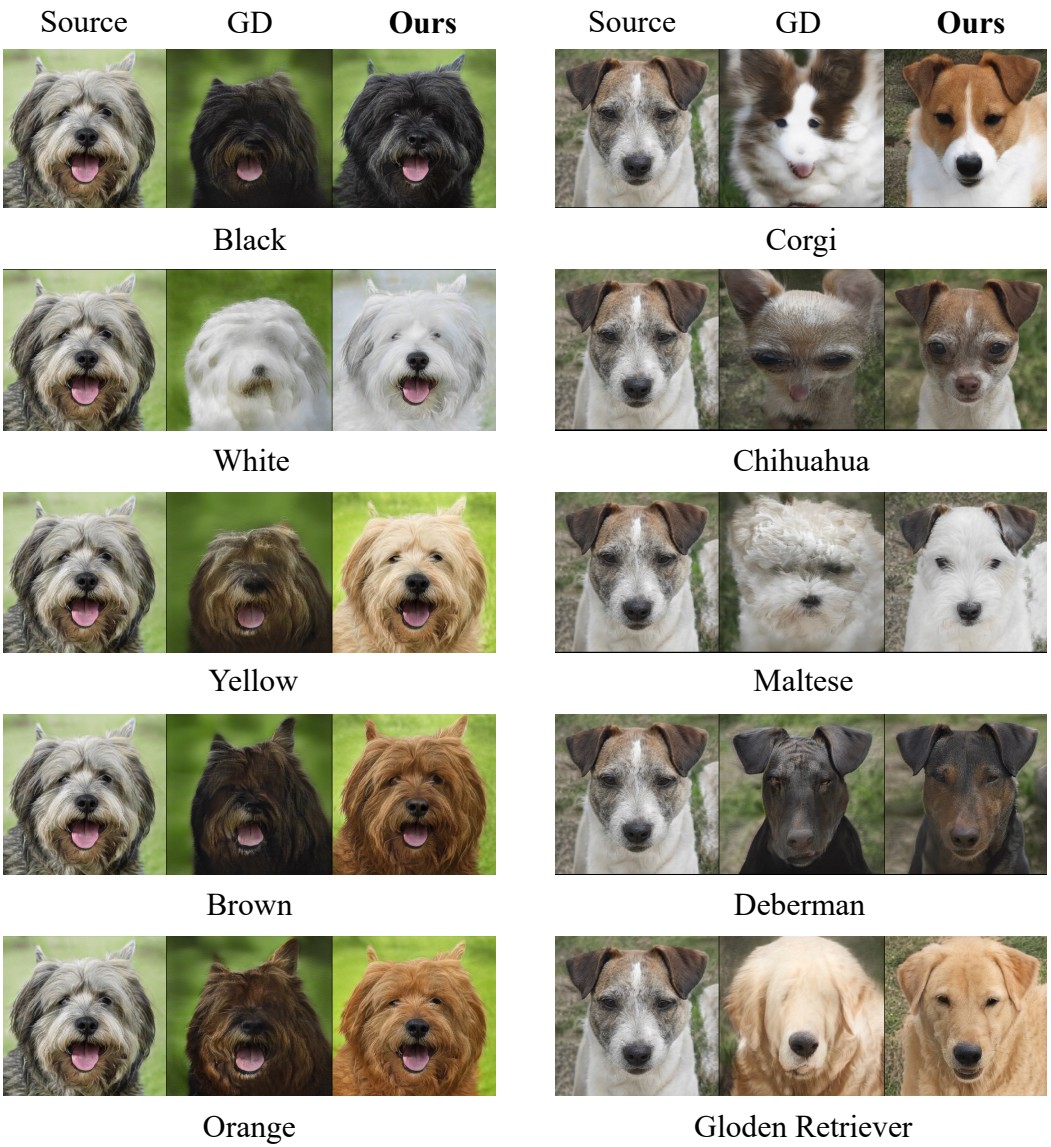

Figure 13: Additional manipulation results on AFHQdog-pretrained StyleGAN-ADA. Equal configuration for both `GlobalDirection` (denoted as GD) and **Multi2One**.

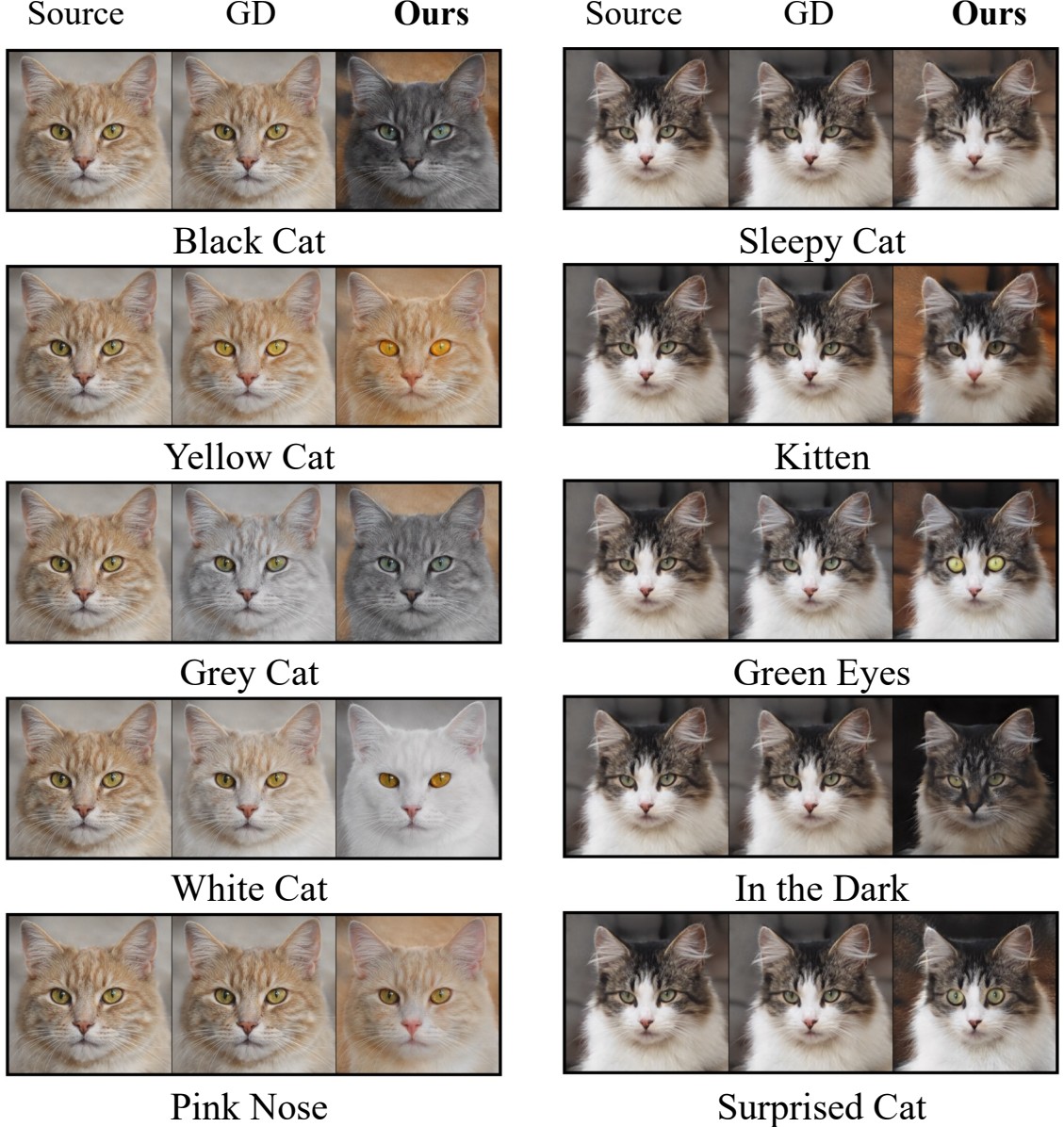

Figure 14: Additional manipulation results on AFHQcat-pretrained StyleGAN-ADA. Equal configuration for both `GlobalDirection` (denoted as GD) and **Multi2One**.

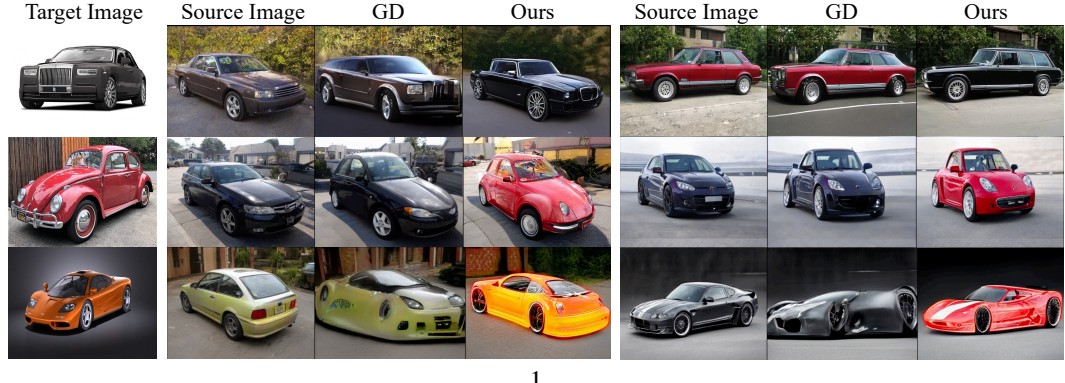

1

Figure 15: Additional manipulation results on LSUNcar-pretrained StyleGAN-ADA. Equal configuration for both `GlobalDirection` (denoted as GD)and **Multi2One**. Used image instead of text as a guidance encoded into CLIP with encoder $E(\cdot)$.

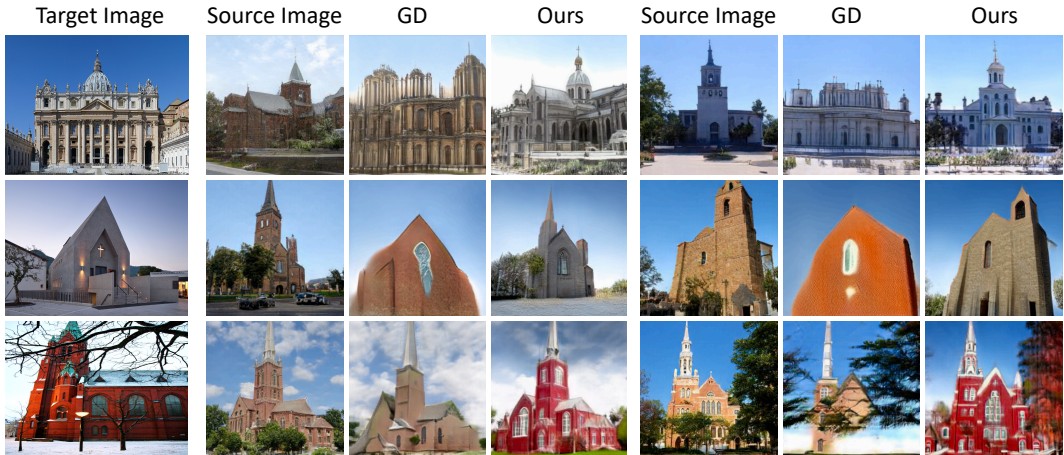

Figure 16: Additional manipulation results on LSUNchuch-pretrained StyleGAN-ADA. Equal configuration for both `GlobalDirection` (denoted as GD) and **Multi2One**. Used image instead of text as a guidance encoded into CLIP with encoder $E(\cdot)$.

# E  EXPERIMENTAL DETAILS

We use unsupervised directions from SeFa and GANspace, both of which are found in intermediate space $\mathcal{W}$ limiting the maximum number of directions to 512, which is the dimension of the intermediate latent space. We use 512 directions from GANSpace and 80 directions from SeFa. Since the earlier layers of StyleGAN mainly shows large structural changes while the deep layers are related to a small detailed changes, we apply the 1024 vectors on three groups of layers: coarse(resolution 4∼32), medium(resolution 64∼128) and fine(resolution 128∼) (Karras et al., 2020a), which amounts to total of 1776 unsupervised directions.

Since the text-guided manipulation directions are found in StyleSpace $\mathcal{S}$, we map the 1776 directions into StyleSpace S, where the maximum value of the style channel parameter in each direction is 15. The unsupervised directions are *dense* vectors, where the elements are non-zeros affecting every channel in StyleSpace. *Dense* manipulation vectors are prone to modify multiple regions of images, causing an image to lose the original identity. Sparsifying the known manipulation vectors $\boldsymbol{\alpha}$ into $\boldsymbol{\alpha}^{(k)}$ where $k = 10, 30, 50$ boosts the disentangled property of the learned dictionary.

We train the dictionary from zero matrix with 30000 epochs. The hyper-parameter $\lambda$ is set as 0.01, and the learning rate is 3.0 with Adadelta (Zeiler, 2012), an optimization method with adaptive learning rate.

# F  COMPARISON WITH OTHER SoTA METHODS

Table 4: Evaluation on other text-guided image manipulation methods.

| Metrics | Methods | | | |
|---|---|---|---|---|
| | **Multi2One** | StyleCLIP GD | StyleCLIP LO | StyleMC |
| $\mathbf{R}_{\text{CLIP}}(t, X_{\text{edited}})$ (↑) | 0.2617 | 0.2482 | **0.2698** | 0.2574 |
| $\mathbf{MSE}(\boldsymbol{\alpha}, \hat{\boldsymbol{\alpha}})$ (↓) | 0.0054 | 0.0244 | **0.0047** | 0.0683 |
| $\mathbf{MSE}(\boldsymbol{\alpha}, \hat{\boldsymbol{\alpha}}\vert_{=0})$ (↓) | **0.0012** | 0.0157 | 0.0018 | 0.0651 |
| $\mathbf{MSE}(\boldsymbol{\alpha}, \hat{\boldsymbol{\alpha}}\vert_{\neq 0})$ (↓) | 0.0041 | 0.0086 | **0.0028** | 0.0032 |
| Inference Time (per image-text) | ✗ | | 100 secs | |

In Tab. 4, we provide additional results from other image manipulation methods, StyleCLIP GD (`GlobalDirection` from Patashnik et al. (2021)), StyleCLIP LO (local optimization from Patashnik et al. (2021)) and StyleMC from (Kocasari et al., 2022). **Multi2One** and StyleCLIP `GlobalDirection` are two unique methods that provides instant manipulation on provided text and image without additional computation for optimization. Local optimization (Patashnik et al., 2021) and StyleMC (Kocasari et al., 2022) are optimization based methods which require about 100 seconds of inference time using a single NVIDIA 2080Ti. Moreover, such optimization based methods do not have disentanglement property unlike **Multi2One** and `GlobalDirection` leading to significantly worse performance in source identity preservation.

The text-image similarity is measured using $\mathbf{R}_{\text{CLIP}}(t, X_{\text{edited}})$ from Eq. (7), which measures the similarity between text $t$ and the manipulated image. We used total of 57 text descriptions on human face[6] to evaluate the performance based on FFHQ-pretrained generator. We manipulate all 6048 channels in StyleSpace accordingly with optimization based methods which do not prune certain channels as a remedy for entanglement. Despite being an input-agnostic manipulation method, **Multi2One** is on par with the input-dependent StyleCLIP Local Optimization method (Patashnik et al., 2021) in the CLIP similarity score, $\mathbf{R}_{\text{CLIP}}(t, X_{\text{edited}})$

The measurement on the ability to find the unsupervised directions $\boldsymbol{\alpha}^{(100)}$ (only the top 100 channels from unsupervised direction $\boldsymbol{\alpha}$ are modified) is measured using mean squared error in Eq. (6). Given

---

[6]The construction of 57 random texts for manipulation on FFHQ pre-trained StyleGAN-ada is described in Appendix A

an $\phi_{\mathrm{CLIP}}(\cdot)$ operation which maps the manipulation effect of $\boldsymbol{\alpha}^{(100)}$, it is desirable to find a direction that only modifies the 100 channels that were non-zero entries among $n$ channels in $\boldsymbol{\alpha}^{(100)}$ while the other $n - 100$ channels are not modified. Therefore, $\mathbf{MSE}\big(\boldsymbol{\alpha},\,\hat{\boldsymbol{\alpha}}|_{=0}\big)$ measure the level of disentanglement which measures the distance between the $n - 100$ channels in $\hat{\boldsymbol{\alpha}}$ and $\boldsymbol{\alpha}^{(100)}$. The result in Tab. 4 shows that ours, **Multi2One**, scored the lowest $\mathbf{MSE}\big(\boldsymbol{\alpha},\,\hat{\boldsymbol{\alpha}}|_{=0}\big)$ proving to be the most disentangled among the four text-guided manipulation methods.

# G ABLATION STUDY ON LOSS TERM

We report the similarity score as $\mathbf{R}_{\mathrm{CLIP}}(t, X_{\mathrm{edited}}) - \mathbf{R}_{\mathrm{CLIP}}(t, X)$, to show the increase in similarity value when modified using the text guidance $t$. Note that $\mathbf{ID}$ loss tends to increase as the image is manipulated properly, therefore it is not a robust metric for evaluating the entanglement level.

Table 5: We report the effect of the hyper-parameter $\lambda$ and the candidates for top-k for creating the sparsified unsupervised directions. $\mathbf{ID}(.,.)$ indicates ArcFace Loss, which measures a change in identity, between the original image and edited image.

|  |  | Similarity | | $\mathbf{ID}(X, X_{edit})$ | |
| --- | --- | --- | --- | --- | --- |
|  | $\lambda$ | 0.1 | 0.01 | 0.1 | 0.01 |
| Top-k | 10 | 0.125 | 0.106 | 0.375 | 0.175 |
|  | 10, 30 | 0.098 | 0.140 | 0.075 | 0.390 |
|  | 10, 30, 50 | 0.115 | **0.141** | 0.417 | 0.284 |
|  | 10, 30, 50, 100 | 0.120 | 0.121 | 0.225 | 0.007 |

In Fig. 17, we visualize the manipulation results to show the effect of the second loss term when removed by $\lambda = 0$ compared to when the second loss term is included by $\lambda = 0.01$. We observe that using the additional loss term does help finding novel directions which are not present in the unsupervised directions, leading to better manipulative ability.

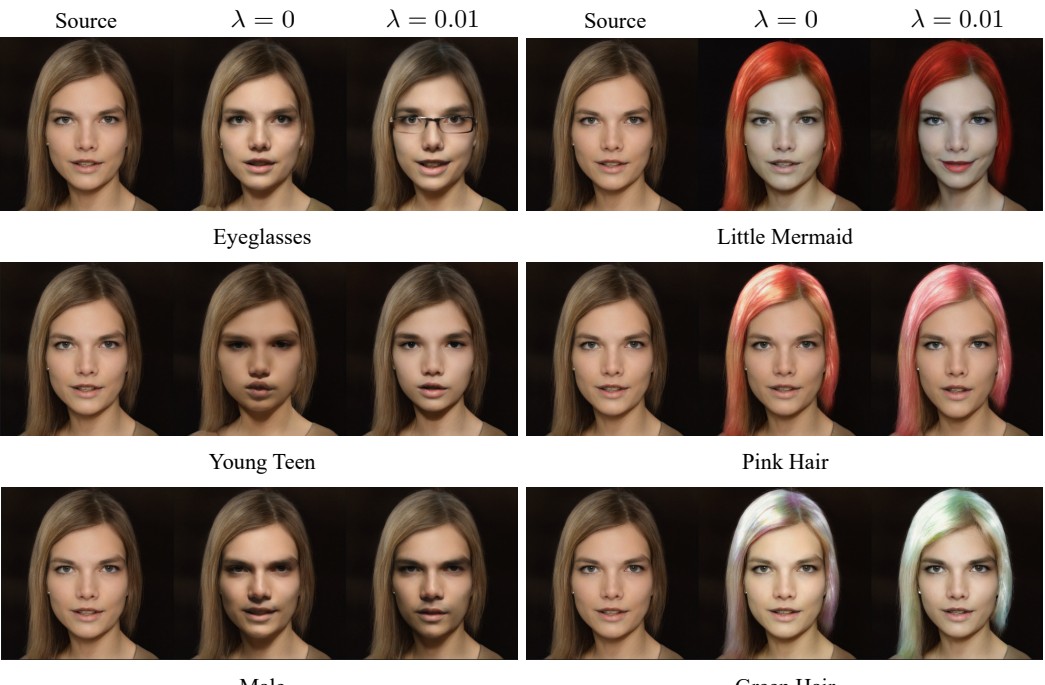

Figure 17: Manipulation results with ablation of the second loss term using $\lambda = 0$ and $\lambda = 0.01$.

# H ABLATION STUDY ON THE EFFECT OF UNSUPERVISED METHODS

The dictionary learning process of **Multi2One** employs the directions $\alpha \in \mathbb{R}^n$ from unsupervised methods (Shen & Zhou, 2021; Härkönen et al., 2020). Therefore, we conduct an ablation study on the effect of using unsupervised directions by comparing the two cases where directions $\alpha$ come from supervised method (Shen et al., 2020) and unsupervised methods (Shen & Zhou, 2021; Härkönen et al., 2020). Fig. 18 shows that using directions from supervised methods are less capable of manipulating images to have desired change. This could also be observed by quantitative measure $\mathbf{R}_{\text{CLIP}}(t, X_{\text{edited}})$ in Tab. 6 since the similarity score between manipulated image and the text guidance is significantly lower using directions from supervised method compared to using unsupervised method. This is due to the limited number of directions that could be found by supervised method, which amounts to only 14 using InterFaceGAN (Shen et al., 2020).

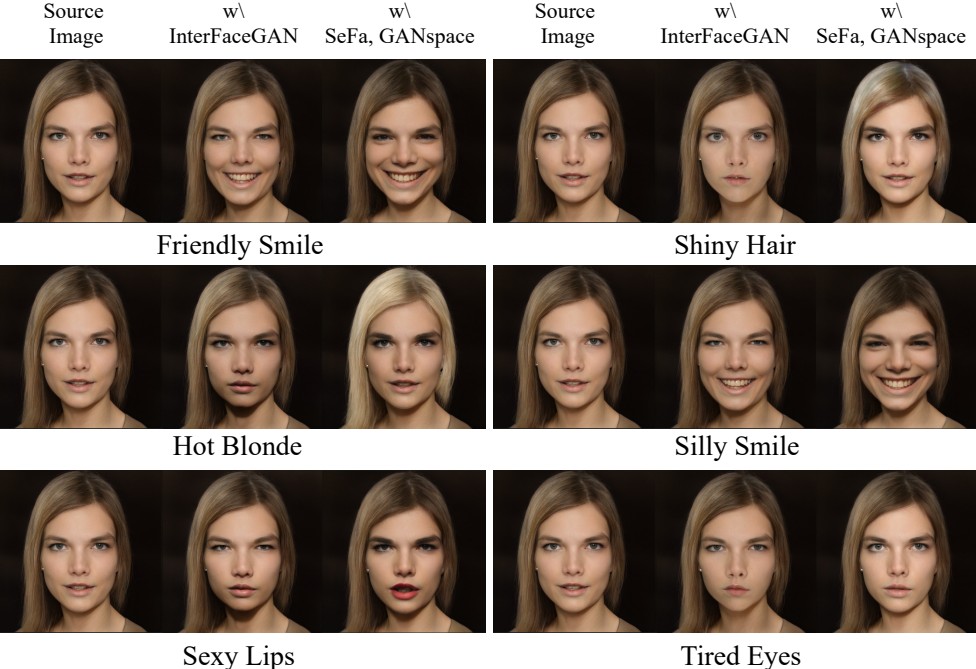

Figure 18: Ablation study on the effect of using unsupervised directions from Härkönen et al. (2020); Shen & Zhou (2021). The substitute the ground truth directions in the dictionary learning process with directions from InterFaceGAN (Shen et al., 2020). We manipulate same number of channels with equal manipulation strength for both.

Table 6: Evaluation on other text-guided image manipulation methods. 100 channels manipulated with manipulation strength of 5. Supervised method indicates using directions from Shen et al. (2020) as *direction*. Unsupervised method indicates that dictionary learning process uses ground truth directions $\alpha$ from Härkönen et al. (2020); Shen & Zhou (2021).

| $\alpha$ | Supervised Method | Unsupervised Method |
|---|---|---|
| $\mathbf{R}_{\text{CLIP}}(t, X_{\text{edited}})$ | 0.228 | **0.254** |

## I  COMPUTATIONAL OVERHEAD

StyleCLIP `GlobalDirection` and **Multi2One** require preprocessing time to compute $\phi_{\mathrm{CLIP}}(\cdot)$ operation. Such operation is computed using either $e_i$ over $n$ channels (StyleCLIP) or $\alpha$ for total of $q$ directions. More specifically, StyleGAN-ADA generator has a fixed StyleSpace dimension consisting of $n$ channels which varies by the resolution of output image. 1024 resolution has 6048 channels, while 512 resolution has 5952 and 256 has 5760. Therefore the total computational time is measured by the number of $\phi_{\mathrm{CLIP}}(\cdot)$ operations calculated over $n$ or $q$ directions using 100 instances of source images to compute the expectation. In Tab. 7 we measure the average computation seconds per iteration using NVIDIA 2080Ti which is equal for both methods. Since one iteration means a single $\phi_{\mathrm{CLIP}}(\cdot)$ operation, comparing the number of iterations shows that our method requires less computation time. Moreover, our method introduces an additional dictionary learning phase which only takes about 15 minutes. Therefore we conclude that our method is slightly more efficient in terms of time to construct the dictionary $D$.

Table 7: Measured time required for a forward pass with batch size 1 in NVIDIA 2080Ti-seconds. SG2 stands for StyleGAN-ADA. The first row show the duration of single iteration in seconds.

| | Preprocessing Time | | | Dictionary Learning |
|---|---|---|---|---|
| Dataset | SG2-**res1024²** | SG2-**res512²** | SG2-**res256²** | All **resolution** |
| Sec/it | 14.98s/it | 13.11s/it | 8.19s/it | |
| GD (iters) | 6048 | 5952 | 5760 | ✗ |
| Ours (iters) | **5328** | **5328** | **5328** | 15 mins |

## J  LIMITATIONS

The flexibility and diversity of text is not fully exerted due to the limited encoding ability and the deterministic representation of CLIP (Radford et al., 2021). For example, it is possible to observe change in pose when using the eigenvectors (vectors that correspond to large eigenvalues often shows large change in image) from unsupervised methods. However, since CLIP is known to have limited ability in terms of understanding the relative position, it is impossible to modify an image using text guidance such as 'move right' and 'turn left'. Moreover, we observe that CLIP encoder fails to represent some of the text instance such as 'Smiling eyes' (Fig. 7) while it is shown in Fig. 1(a) that change in eyes happens using text 'Smile'. Since it is proven that the editing direction of 'smiling eyes' could be discovered when provided with appropriate text, we conjecture that the CLIP encoder fails to represent some text input faithfully leading to unsuccessful manipulation results.

