# OpenReview forum: "Learning Input-agnostic Manipulation Directions in StyleGAN with Text Guidance"
_ICLR.cc/2023/Conference — ICLR 2023 poster_

### Official Review · Reviewer_jy1p · 2022-10-19

**Confidence:** 2
**Correctness:** 3
**Technical Novelty And Significance:** 3
**Empirical Novelty And Significance:** 2
**Recommendation:** 6

**Clarity, Quality, Novelty And Reproducibility:**

This motivation of this paper is clear and the idea to consider the inter-channel correlations is novel to me. About the reproducibility, the authors promise to release the code.

**Strength And Weaknesses:**

**Strengths:**
+ The idea to consider the inter-channel correlations during latent editing is intuitive and reasonable. The paper is well motivated.
+ In general, the visual and editing quality of the proposed method is significantly better than the baseline method.

**Weaknesses:**
+ The idea of the second loss term in Eq. (4) is hard to understand to me. What is the intuitive motivation or physical meaning behind this loss term? Is there any visual results with and without this loss term to help readers better understand its effect?
+ In ablation study, the effect of second loss term is better to be validated by setting lambda=0. Current, only lambda=0.1 and 0.01 is compared quantitatively. Visual results are suggested to be given for better comparison.
+ The limitation is suggested to be discussed. For example, why in Fig. 11, the text is smiling eyes, but the eyes have nearly no changes while the mouth changes a lot.
+ The conclusion is suggested to be given.

**Some small issues:**
+ The text in Fig. 6 is too small.


**Summary Of The Paper:**

This paper proposes to improve the GlobalDirection method of StyleCLIP to further consider the inter-channel correlations when finding the editing vectors in StyleSpace. A novel dictionary learning loss is proposed to learn such correlations and experiment validates the effectiveness of the method compared to the StyleCLIP.

**Summary Of The Review:**

This paper proposes to improve the GlobalDirection method of StyleCLIP to further consider the inter-channel correlations when finding the editing vector. I can generally understand the main idea of dictionary learning but cannot fully nail down the meaning and function of the second loss term. From the experiment results, it seems that the proposed dictionary learning is effective to improve the editing quality.

---

> ### Author Response · Authors · 2022-11-17
> **Response to Reviewer jy1p**
>
> We sincerely appreciate your time and effort in reviewing our paper.
> We would like to address the questions below.
>
> ### Q1
> > The idea of the second loss term in Eq. (4) is hard to understand to me. What is the intuitive motivation or physical meaning behind this loss term? Is there any visual results with and without this loss term to help readers better understand its effect?
>
> The second term updates the dictionary $D \in \mathbb R^{p \times n}$ to discover wider variety of directions which are not present in unsupervised directions $\boldsymbol \alpha \in \mathbb R^n$ that our method is using as a training data.
>
> As clarified in the revision, we use the sparsified augmentation, $\boldsymbol \alpha^{(k)}$, of unsupervised directions $\boldsymbol \alpha$. By using the augmented directions for our second term, we can enforce the dictionary to learn new and disentangled directions that are not provided by unsupervised methods.
>
> For more details, please refer to the revised Section 4. We also have illustrated the intuition behind the second loss term on the concept figure (Figure 4) using a specific example.
>
> ***
> ### Q2
> > In ablation study, the effect of second loss term is better to be validated by setting lambda=0. Current, only lambda=0.1 and 0.01 is compared quantitatively. Visual results are suggested to be given for better comparison.
>
> |  | $\lambda=0$ | $\lambda=0.01$ |
> |---|---|---|
> | 10 | 0.235 | **0.242** |
> | 30 | 0.240 | **0.249** |
> | 50 | 0.243 | **0.251** |
> | 100 | 0.248 | **0.254** |
>
> In the table above, we provide the ablation result for the second term with the similarity metric $\mathbf R_\mathrm{CLIP}(t, X_\mathrm{edited}) (\uparrow)$. This proves that using the second loss term with $\lambda=0.01$ yields better manipulation quality compared to ours with $\lambda=0$. We have added the ablation result in Table 3 of the revised paper.
>
> We provide additional visualization on the effect of the second loss term in Figure 17. As shown in this figure, we observe that the dictionary trained with $\lambda=0.01$ is better in terms of the manipulation quality given the text.
>
> ***
> ### Q3
> > The limitation is suggested to be discussed. For example, why in Fig. 11, the text is smiling eyes, but the eyes have nearly no changes while the mouth changes a lot. The conclusion is suggested to be given.
>
> - Upon the request of the reviewer, we have added limitation in Appendix J. We quote the limitations from the revised paper here:
>
> "The flexibility and diversity of text is not fully exerted due to the limited encoding ability and the deterministic representation of CLIP. For example, it is possible to observe change in pose when using the eigenvectors (vectors that correspond to large eigenvalues often show large change in image) from unsupervised methods. However, since CLIP is known to have limited ability in terms of understanding the relative position, it is impossible to modify an image using text guidance such as 'move right' and 'turn left'. Moreover, we observe that CLIP encoder fails to represent some of the text instance such as 'Smiling eyes' (see Figure 8) while it is shown in Figure 1(a) that such editing is possible using another text guidance 'Smile'. Since it is proven that the editing direction of 'smiling eyes' could be discovered when provided with appropriate text, we conjecture that the CLIP encoder fails to represent some text input faithfully, leading to unsuccessful manipulation results."
>
> - We have also included the conclusion section, which we also quote below:
>
> "Text-guided image manipulation has been largely preferred over unsupervised methods in finding manipulation directions due to its ability to find wide variety of directions corresponding to flexible text input. However, we have investigated that the state-of-the-art method of text-guided image manipulation has limited coverage over possible editing directions using StyleGAN. In this paper, we prove that the problem rises from a simple assumption that a single-channel manipulation could fully represent the manipulation ability of StyleGAN. To overcome this issue, we have proposed a dictionary learning framework that embeds the interactive and collective editing effect based on modifying multiple style channels into CLIP space jointly. Our method have been proven to be superior over other text-guided image manipulation methods in terms of manipulative ability and disentanglement."

---

> > ### Comment · Reviewer_jy1p · 2022-12-07
> > **Thank you for the response**
> >
> > Dear authors,
> >
> > I thank the authors for their response.
> > The authors have fixed or added new contents to the earlier version of the paper and addressed my concerns.
> > After reading other reviewer comments and the authors' response, I am still positive towards this paper.

---

### Official Review · Reviewer_bYKo · 2022-10-23

**Confidence:** 4
**Correctness:** 2
**Technical Novelty And Significance:** 2
**Empirical Novelty And Significance:** 2
**Recommendation:** 6

**Clarity, Quality, Novelty And Reproducibility:**

In general, this submission is well-written and easy to follow. It has good results compared to chosen baselines. However, as for me, it suffers from limited technical novelty and still lacks some experimental results. Since it would be tricky to train networks from scratch, codes are welcome for reproducibility.

**Strength And Weaknesses:**

Strengths:

+ The authors design experiments to support their motivations by showing that the GlobalDirection can not provide consistent embeddings. I like these experiments, which makes the statement clear.
+ The shown visual results look pleasing and demonstrate some improvement over GlobalDirection.
+ The writing is clear and easy to follow.

Weakness:

+ The proposed approach is an incremental improvement upon StyleCLIP and GlobalDirection methods. Specifically, this submission can be regarded as a straightforward extension of GlobalDirection by extending the single-channel editing direction to multi-channel editing direction. Also, the idea of discovering the editing direction and training the dictionary also borrows the results of unsupervised latent direction discovery methods such as SeFA and GANSpace. This makes me feel like this submission might suffer from limited novelty.

+ As mentioned in Section E, the proposed method uses the output of previous methods (e.g. SeFA and GANSpace). So the authors need an ablation study (only use multi-channel learning strategy) along with corresponding discussions to rule out the influence of this point and then highlight the influence of multi-channel modeling. And the fact that this method depending on SeFA and GANSpace's outputs, also might make it suffer from similar limitations to SeFA and GANSpace.
+ How to evaluate the image synthesis results is still an open problem. Thus this submission mainly depends on the qualitative comparisons, which might be very subjective. Although this submission presents some basic quantitative comparisons, I suggest to use some widely used quantitative metrics, like Inception score or more advanced alternatives, to evaluate the generation results instead of MSE and Cosine similarity. At least, the user studies are needed for evaluations over randomly generated results.
+ In this submission, the authors only compare with the original StyleCLIP but there are other variants. So more empirical comparisons need to be conducted with more SOTA methods.
+ While the proposed method shows better results over chosen baselines, I am curious about the comparisons in terms of computation overhead and other additional limitations introduced by multi-channel modeling.

**Summary Of The Paper:**

In this paper, the authors propose a dictionary learning framework to improve StyleCLIP, one state-of-the-art image manipulation method via text guidance. Compared to StyleCLIP which operates on one single channel independently, this method tries to embed the multiple style channels into CLIP space jointly. Some experiments are performed to support the advantages of the proposed method.

**Summary Of The Review:**

This submission details how it extends from GlobalDirection to achieve multiple channel modeling. However, this motivation seems to be somewhat trivial and just a straightforward revision, even given its superior results. The authors can also improve their paper by adding more experimental results to support their claims. But I would not like to object acceptance if other reviewers prefer to do so.

---

> ### Author Response · Authors · 2022-11-17
> **Response to Reviewer bYKo (3/3)**
>
> ### Q5
> > While the proposed method shows better results over chosen baselines, I am curious about the comparisons in terms of computation overhead and other additional limitations introduced by multi-channel modeling.
>
> Both $\mathtt{GlobalDirection}$ and ours allow instant manipulation based on the preprocessed dictionary $D$. The major computational overhead comes from the construction of this dictionary $D$. The difference in terms of computation for both methods comes from the number of computing $\boldsymbol \phi_{\mathrm{CLIP}}(\cdot)$ and the existence of dictionary optimization process.
>
> The pre-trained FFHQ generator has 6048 channels in StyleSpace $\mathcal S$ and computing $\boldsymbol \phi_{\mathrm{CLIP}}(\cdot)$ for a single channel  takes 14.98 seconds, therefore the total preprocessing time for $\mathtt{GlobalDirection}$ is roughly 25 hours. On the other hand, our method is trained over 5328 directions by unsupervised methods, hence it consumes about 22 hours for preprocessing.
>
> Our proposed method requires an additional optimization stage but it only takes 15 minutes with a single NVIDIA 2080Ti.
>
> In short, our method needs less computation time thus no additional overhead is introduced when using our method.
>
>
> ***
> ### Q6
> > Since it would be tricky to train networks from scratch, codes are welcome for reproducibility.
>
> The authors promise to release the code for reproducibility in the near future.

---

> > ### Comment · Reviewer_bYKo · 2022-12-07
> > **Thanks for your responses**
> >
> > Dear authors,
> >
> > Thank you for your detailed responses to my questions. I think the rebuttal has already addressed most of the issues with extensive clarifications and experiment results. And the remaining ones cannot be significantly improved without re-designing the framework from scratch. If the authors promise to release their codes with detailed instructions to the experimental settings for full reproduction, I will not object to accept. So I raised my score to 6.

---

> > > ### Author Response · Authors · 2022-12-07
> > > **Thank you for the response**
> > >
> > > We are happy to know that our response have addressed most of the issues. We greatly appreciate your time and effort in giving constructive comments on the paper. We promise again to release the codes with details for experimental settings. Thank you for the positive feedback.

---

> ### Author Response · Authors · 2022-11-17
> **Response to Reviewer bYKo (2/3)**
>
>
> * * *
> ### Q3
> > How to evaluate the image synthesis results is still an open problem. Thus this submission mainly depends on the qualitative comparisons, which might be very subjective. Although this submission presents some basic quantitative comparisons, I suggest to use some widely used quantitative metrics, like Inception score or more advanced alternatives, to evaluate the generation results instead of MSE and Cosine similarity. At least, the user studies are needed for evaluations over randomly generated results.
>
> Evaluation on text-guided image manipulation do not have a robust metric for quantitatively evaluating the performance. Note that Inception score or related metrics may be used for assessing the quality of image itself, but they cannot be employed to measure how well the edited image matches the input text. Even StyleCLIP did not report any quantitative metrics except for one of the three methods, global mapper, which measures cosine similarity between text and image for a single text guidance only. Likewise, StyleMC and TediGAN also did not report quantitative score for there is no standard quantitative measure in assessing the manipulation quality and the level of disentanglement given text input.
>
> Though we have designed two metrics to objectively assess the performance of text-guided image manipulation methods, we do agree on the necessity of more widely used metrics. Therefore, as the reviewer suggests, we are conducting the user studies over randomly generated results. Looking at the many results in appendix where we did not specifically screen the input texts for us, we are confident that even human evaluation will also validate the excellent performance of our method (as all other reviewers acknowledged). However, since it takes time to secure the text set for the fairest verification and to prepare the results for them, we cannot include it in the revision of this first discussion phase. We will definitely include it in the final version.
> * * *
>
> ### Q4
> > In this submission, the authors only compare with the original StyleCLIP but there are other variants. So more empirical comparisons need to be conducted with more SOTA methods.
>
> As discussed in the introduction, our main focus is the *instant* manipulation for the given text in the inference time, considering usability in practice. Our method and StyleCLIP $\mathtt{GlobalDirection}$ are two unique methods under this category.
>
> However, we provide additional results from other image manipulation methods, StyleCLIP GD, StyleCLIP LO (Local Optimization from StyleCLIP), and StyleMC. Please refer to Appendix F for more details in this experiment.
>
> The table below shows that StyleCLIP Local Optimization scored the highest CLIP similarity score $\mathbf R_{\mathrm{CLIP}}(t, X_{\mathrm{edited}})$. Despite being an input-agnostic method, our method shows high similarity score on par with the input-dependent method.
>
> We also demonstrate the Mean Squared Error metrics described in Equation 6. We emphasize that our method shows outstanding disentanglement ability with the lowest $\mathbf{MSE}(\boldsymbol \alpha, \hat{\boldsymbol \alpha}\cdot \mathbb I_{\{0\}}(\boldsymbol \alpha))$.
>
>
> |  | Ours \| StyleCLIP GD | StyleCLIP LO \| StyleMC |  |
> |---|:---:|:---:|---|
> | $\mathbf R_{\mathrm{CLIP}}(t, X_{\mathrm{edited}})$ (&#8593;) | **0.2617** \| 0.2482 | ***0.2698*** \| 0.2574 |
> | $\mathbf{MSE}(\boldsymbol \alpha, \hat{\boldsymbol \alpha})$ (&#8595;) | **0.0054** \| 0.0244 | ***0.0047*** \| 0.0683 |
> | $\mathbf{MSE}(\boldsymbol \alpha, \hat{\boldsymbol \alpha}\cdot \mathbb I_{\{0\}}(\boldsymbol \alpha))$ ( &#8595; ) | ***0.0012*** \| 0.0157 | **0.0018** \| 0.0651
> | $\mathbf{MSE}(\boldsymbol \alpha, \hat{\boldsymbol \alpha}\cdot \mathbb I_{\mathbb R\neq0}(\boldsymbol \alpha))$ ( &#8595; ) | **0.0041** \| 0.086 | **0.0028** \| 0.0032 |
>
> ***

---

> ### Author Response · Authors · 2022-11-17
> **Response to Reviewer bYKo (1/3)**
>
> We greatly appreciate the reviewer for the constructive comments on our paper.
>
> ### Q1
> > The proposed approach is an incremental improvement upon StyleCLIP and GlobalDirection methods. Specifically, this submission can be regarded as a straightforward extension of GlobalDirection by extending the single-channel editing direction to multi-channel editing direction. Also, the idea of discovering the editing direction and training the dictionary also borrows the results of unsupervised latent direction discovery methods such as SeFa and GANSpace. This makes me feel like this submission might suffer from limited novelty.
>
> It is true that our method is built upon the general framework of $\mathtt{GlobalDirection}$ (GD) and in the process of learning the dictionary, we use directions from unsupervised methods. However, unlike what the reviewer said, it is very challenging to extend the single-channel based GD to multi-channel based manipulation. Considering all possible combinations of multi-channel manipulations, it will be easily computationally intractable (exponentially many cases for the number of channels, which amounts to 6048).
>
> To solve this issue, we are learning a dictionary that connects StyleSpace and CLIP space for text. At this time, since there is no paired ground truth for dictionary learning, we propose to indirectly utilize the directions found by unsupervised methods. Especially, in order to minimize the limitations of using already known directions by unsupervised methods as a training data, we propose a non-trivial objective and verify its excellence via extensive experiments. Note that our dictionary learning means more than simply reproducing the known directions derived by unsupervised methods. As the dictionary learns the relationship between channels in StyleSpace and CLIP space, we can find manipulations that could not be found with unsupervised methods using diverse text inputs. Please see Figure 7 - our method can perform quite well for new text input where unsupervised methods (or $\alpha$ used for our training data) fail. We revised Section 1, Section 3.2 and Section 4 to make these points more evident.
>
>
> ***
>
> ### Q2.
> > As mentioned in Section E, the proposed method uses the output of previous methods (e.g. SeFA and GANSpace). So the authors need an ablation study (only use multi-channel learning strategy) along with corresponding discussions to rule out the influence of this point and then highlight the influence of multi-channel modeling. And the fact that this method depending on SeFA and GANSpace's outputs, also might make it suffer from similar limitations to SeFA and GANSpace.
>
>
> - **Regarding ablation study:**
>
>   As mentioned above, there is no way to remove the effect of the direction found by the unsupervised method and purely check the effect of multi-channel due to the complexity issue (even if we naively extend Global Direction and manipulate only two channels simultaneously, the effect of which will be not much different from single channel based way, we should consider a total of ${6048 \choose 2}$ directions.)
>
>   It may not have been the experiment the reviewer intended, but instead we conducted an additional ablation study using the direction found by the supervised method, namely InterFaceGAN [1], instead of the unsupervised method. Please refer to Figure 18 and Table 6 in Appendix H.
>
>   [1] Shen, Y., Yang, C., Tang, X., & Zhou, B. (2020). Interfacegan: Interpreting the disentangled face representation learned by gans. IEEE transactions on pattern analysis and machine intelligence.
>
> - **Suffers from similar limitations to SeFA and GANSpace?**
>
>   Also, as mentioned above, we only use the directions found by unsupervised methods as training data to learn the relationship between StyleSpace and CLIP space, and hence do not inherit their innate disadvantages.
>
>   Our method is not simply reconstructing the directions of unsupervised methods. Our method allows manipulations that are not possible with unsupervised directions with new text (see Figure 7), and is much more disentangled than them (See Figure 9 of Appendix C to understand how entangled the directions by unsupervised methods are).

---

### Official Review · Reviewer_RdgP · 2022-10-24

**Confidence:** 4
**Correctness:** 3
**Technical Novelty And Significance:** 3
**Empirical Novelty And Significance:** 3
**Recommendation:** 6

**Clarity, Quality, Novelty And Reproducibility:**

The overall quality of the paper is pretty good, but some formulations could be made more clear (such as eq.2) and some figures could be further improved (such as fig.1 – currently there is too much text thus it looks messy).

**Strength And Weaknesses:**

Strength:
1. Global direction editing is charming since it can be transferred to different new images without local optimization. An effective method around text-guided editing is a promising task.
2. The analysis is interesting. Especially the experiment is impressive, i.e., single-channel editing provides no change while joint-channel editing can change the global shape. A corresponding method is derived based on the observation, which makes it more convincing. The results look good compared with the original StyleCLIP dictionary. The joint-channel editing may provide other unsupervised editing work with several insights (the collective and interactive relation among multiple channels should be considered).
3. The authors conduct adequate experiments on sufficient datasets to prove the manipulation performance of the proposed method surpasses StyleCLIP.

Weakness:
1. One question, beyond text-guided editing, is it possible to use such a global direction to provide a robust editing method on the large pose or expression change? Existing methods have attempted to use 3DMM representation or other unsupervised editing directions to inject into 2D StyleGAN, but the results are not that robust.
2. Some figures are pretty messy. For instance, there are too many texts of various fonts/sizes/colors in Fig.1 and Fig.4. Please consider unifying some of them.
3. Some terms lacked explanation when first mentioned, such as △s and e(i) in Eq(2), easily confusing the reader.
4. The process of StyleCLIP’s obtaining a dictionary should be represented more clearly in case the reader is unfamiliar.

**Summary Of The Paper:**

This paper proposes a new method for image manipulation with ‘global direction’, which is case-agnostic and applicable to all input images. The authors argue that the previous work StyleCLIP provides a dictionary to quickly find out the channel-wise manipulation while the collective and interactive relation among multiple channels is not considered. Thus the manipulation performance is limited. The authors propose a new manipulation method by considering the manipulation effect coming from the interaction with multiple other channels.


**Summary Of The Review:**

The reviewer recommends a weak accept of the paper and is pretty confident.

---

> ### Author Response · Authors · 2022-11-17
> **Response to Reviewer RdgP**
>
>
> We thank the reviewer for the appreciation on our paper and the valuable comments. Hereby we address the concerns and questions from the reviewer.
>
> ### Q1
> > One question, beyond text-guided editing, is it possible to use such a global direction to provide a robust editing method on the large pose or expression change? Existing methods have attempted to use 3DMM representation or other unsupervised editing directions to inject into 2D StyleGAN, but the results are not that robust.
>
> Modification on pose seems to be impossible due to the limited encoding ability of CLIP. Analysis on the ability of CLIP has revealed that such cross-modal contrastive learning approach cannot encode relative positions, therefore guiding a StyleGAN generator to convert the given object to move positions or pose seems impossible. We have tried to manipulate the source image to show change in pose but were unable to produce any kinds of pose change on StyleCLIP Local optimization, StyleCLIP Global direction, TediGAN, StyleMC and our method. On the other hand, we observe that large change in expression is possible. Should the reviewer request examples on large change in expression, we would be happy to provide the manipulation results.
>
> ***
> ### Q2
> > Some figures are pretty messy. For instance, there are too many texts of various fonts/sizes/colors in Fig.1 and Fig.4. Please consider unifying some of them.
>
> We have completely revised figure 1 which depicts the overall framework (Figure 2 in the revised version) and figure 4 which serves as a concept diagram of our method. As the reviewer requested, we have unified all the fonts, sizes and colors in figures.
>
> > Some terms lacked explanation when first mentioned, such as △s and e(i) in Eq(2), easily confusing the reader.
>
> We have carefully revised section 3 and section 4 and ensured that there are no notations without explanation.
>
> In the original version, we denoted the single-channel manipulation vector, $\Delta \boldsymbol s_i\in \mathbb R^n$, which is defined as a zero-vector except for the $i$-th entry as in $\sigma_i \boldsymbol e^{(i)}$. Here, $\boldsymbol e^{(i)}\in \mathbb R^n$ is a standard basis vector whose $i$-th entry is 1 and the others are all zeros.
>
> To avoid confusion, we have simplified the notation of such single-channel manipulation into $\boldsymbol e_i\in \mathbb R^n$ in the revised version of the paper.
>
> ***
>
> ### Q3
> > The process of StyleCLIP’s obtaining a dictionary should be represented more clearly in case the reader is unfamiliar.
>
> Based on the reviewer's comment, we have added clear explanation to help understand StyleCLIP $\mathtt{GlobalDirection}$ in Section 3.1.

---

### Official Review · Reviewer_v5mm · 2022-10-25

**Confidence:** 3
**Correctness:** 2
**Technical Novelty And Significance:** 2
**Empirical Novelty And Significance:** 2
**Recommendation:** 6

**Clarity, Quality, Novelty And Reproducibility:**

*Clarity*: The paper is not well-written, and is really hard to follow.

*Quality and Novelty*: It is hard to determine the quality and novelty of this work since the motivation and proposed framework is not clearly introduced in the paper.

*Reproducibility*: Although the proposed method is not well described, the author promise to release the source code in the future.

**Strength And Weaknesses:**

*Strength*
1. The authors promise to release the source code in the future, which can stimulate further research in this field.
2. From the quantitative and qualitative results, the proposed method seems to have favorable editing capability compared to the state-of-the-art methods.

*Weakness*
1. The paper is not well written. Particularly, Section 3 and 4 is hard to follow. The syntaxes in the figure and the text description are not alighted. For example, $\Delta{\mathcal{S}}$ is used in Figure 1, but in the text descriptions it seems to be $\Delta{s}$. There are many confusing and unclear descriptions as well. For instance, in Section 3.2, what do the authors mean by "a large portion of the instances from unsupervised directions"? What do the "instance" and "directions" refer to? Moreover, what does "represent new directions to the fullest capacity" mean? The notation $\alpha$ is also not properly defined. What is $alpha$? What is the dimensionality? For section 4, how do the authors use the sparsified $\alpha^{(r)}$ in Equation 4?
2. The experiments shown in Section Table 1 are also confusing. For Table 1, how is the direction term $\alpha$ computed for the GlobalDirection method computed?
3. As the goal of this paper is to edit images using text, what is the importance of recovering the unsupervised directions in this paper?


**Summary Of The Paper:**

This paper works on leveraging the StyleGAN model for text-guided image manipulation. Starting from latent directions discovered by the unsupervised method, the authors propose an approach, with a sparsify trick, to further disentangle the editing directions. Experiments conducted on the FFHQ, LSUN, AFHQ datasets show that the proposed method performs favorably against the state-of-the-art method.

**Summary Of The Review:**

I suggest the authors to carefully revise the manuscript, especially Section 2 and 3. Although favorable quantitative and qualitative results are shown, the motivation, intuition, technique, and novelty behind this paper is not well-demonstrated.

--- post rebuttal ---
The authors have revised the manuscript, and now Section 2 and 3 are more clear, so I'm not against to accept this work.

---

> ### Author Response · Authors · 2022-11-17
> **Response to Reviewer v5mm**
>
>
> We thank the reviewer for the valuable and constructive suggestions. As the reviewer suggested, we have thoroughly revised the paper carefully with focus on Section 3 and Section 4. We believe that representation quality of our submission has been greatly improved and our method including motivation and novelty is now clearly demonstrated in the revision.
>
> ---
> ### Q1.
>
> Through revision, we improved readability by redrawing figure 2, and Figure 4 including concept figure. The confusing notation was readjusted, and there was no mismatch.
>
> > In Section 3.2, what do the authors mean by "a large portion of the instances from unsupervised directions"? What do the "instance" and "directions" refer to?
>
> We apologize for the ambiguity in this sentence. We clarify that "a large portion of the instances from unsupervised directions" means that there are many editing directions among the 1024 unsupervised directions (512 directions discovered by GANspace [1] and another 512 directions by SeFa [2]) that the state-of-the-art method, StyleCLIP $\mathtt{GlobalDirection}$ cannot recover. Instance means a single direction out of 1024 directions discovered by GANspace and SeFa. We clarified this in the revision.
>
> > What does "represent new directions to the fullest capacity" mean?
>
> Sorry for the confusion. We originally meant that $\mathtt{GlobalDirection}$ fails to find new directions given text input even when pre-trained StyleGAN actually can generate such manipulations. Here, `new directions' means the manipulation directions that have not been discovered by supervised and unsupervised methods. In order not to cause any misunderstanding, the part has been deleted. Instead, we demonstrate through the Figure 1(b) that $\mathtt{GlobalDirection}$ cannot find a direction corresponding to the given random text while such directions do exist in pre-trained StyleGAN (ours can find such directions).
>
>
> > The notation $\boldsymbol \alpha$ is also not properly defined and its dimensionality? For section 4, how do the authors use the sparsified $\boldsymbol \alpha^{(k)}$ in Equation 4?
>
> The notation $\boldsymbol \alpha$ represents a direction found by unsupervised methods. In the revision, we defined this (with its dimensionality) in Section 3.2 and we reminded of this when necessary throughout the paper. (as mentioned above, we resolve confusion about notation throughout the paper in the revision).
>
>
>
> ***
>
> ### Q2.
> >The experiments shown in Section Table 1 are also confusing. For Table 1, how is the direction term computed for the GlobalDirection method computed?
>
> In Table 1, we show that GlobalDirection
> cannot effectively recover the directions found by unsupervised methods. But, since there is no text for $\boldsymbol \alpha$, for a given direction $\boldsymbol \alpha$ by unsupervised methods, we define $\boldsymbol \phi_{\mathrm{CLIP}}(\boldsymbol \alpha)=\mathbb E_{\boldsymbol{s}\in \mathcal S}[\mathrm{CLIP}\big(G_s(\boldsymbol{s} + \boldsymbol \alpha)\big) - \mathrm{CLIP}\big(G_s(\boldsymbol{s} - \boldsymbol \alpha)\big)]$, which encodes the direction $\boldsymbol \alpha$ into the CLIP embedding space as $\texttt{GlobalDirection}$ does for a single channel. We use this $\boldsymbol \phi_{\mathrm{CLIP}}(\boldsymbol \alpha)$ instead of $\mathrm{CLIP}(t)$. Since the caption on Table 1 which says 'with text $t$' might be misleading, we clarify again that the text guidance $t$ is replaced with $\boldsymbol \phi_{\mathrm{CLIP}}(\boldsymbol \alpha)$.  You can find more details on this in the revision.
>
> * * *
> ### Q3.
> > As the goal of this paper is to edit images using text, what is the importance of recovering the unsupervised directions in this paper?
>
> As the reviewer commented, the goal of this paper is to find a manipulation direction that corresponds to the given text. However, here since there is no paired ground truth of text and manipulation direction corresponding to the text, we embed the directions found by existing unsupervised methods into the CLIP space and learn a dictionary to reproduce them in the CLIP space. Note that this has more meaning than simply reproducing the known directions derived by unsupervised methods. As the dictionary learns the relationship between channels in StyleSpace and CLIP space, we can find manipulations that could not be found with unsupervised methods using diverse text inputs. Please see Figure 7 - our method can perform quite well for new text input  where unsupervised methods (or $\boldsymbol\alpha$ used for our training data) fail.
>
> Also note that the second term in our objective help to have disentangled directions please kindly refer to Section 4 from "To help close the gap between Eq.(3) and Eq.(4)".

---

> > ### Comment · Reviewer_v5mm · 2022-12-07
> > **Response**
> >
> > Thanks for the responses. The authors have revised the manuscript and now Section 2 and 3 are more clear.

---

> > > ### Author Response · Authors · 2022-12-07
> > > **Thank you for the response**
> > >
> > > We thank the reviewer for kindly responding and raising the rate. All your comments on the paper have been greatly constructive and the authors greatly appreciate it.

---

### Decision · Program_Chairs · 2023-01-20

**Decision:**

Accept: poster

**Justification For Why Not Higher Score:**

Even though all the reviewers are positive towards this paper, they were not very excited about the method and gave relatively low novelty scores.

**Justification For Why Not Lower Score:**

The AC and the reviewers feel that this work is a solid contribution to the community and would like to see it in the conference if space allows.

**Metareview: Summary, Strengths And Weaknesses:**

This paper proposes a new dictionary learning method to improve StyleCLIP for text-guided image manipulation, which considers the manipulation effect coming from the interaction among multiple channels. The analysis is insightful, which diagnoses the manipulation effect of StyleCLIP. The proposed method is novel and its effectiveness is shown by both quantitative and qualitative experimental results on multiple datasets. Most of the concerns raised by the reviewers, e.g., lack of clarity,  have been addressed by the rebuttal and revision, and the reviewers are positive towards this paper. Overall, the AC acknowledges the contributions of this work and would like to see it in the conference if space allows.

**Note From Pc:**

if the above contains the word "oral" or "spotlight" please see: "oral" presentation means -> notable-top-5% and "spotlight" means -> notable-top-25%. As stated in our emails, we are disassociating presentation type from AC recommendations

**Summary Of Ac-Reviewer Meeting:**

The AC tried to schedule a meeting for discussion but was unable to because of the unavailability of the reviewers and the AC. But the reviewers actively updated their reviews (some of them emailed the AC about their most recent opinions of this paper). Overall, the reviewers are inclined to an acceptance of this paper.